# Hemispherotomy leads to persistent sleep-like slow waves in the isolated cortex of awake humans

Michele Angelo Colombo[1⊕], Jacopo Favaro[2⊕], Ezequiel Mikulan[3], Andrea Pigorini[4,5], Flavia Maria Zauli[1,6,7], Ivana Sartori[7], Piergiorgio d'Orio[7,8], Laura Castana[7], Irene Toldo[2], Stefano Sartori[2,9,10], Simone Sarasso[1], Tim Bayne[11,12,13], Anil K. Seth[12,14,15], Marcello Massimini[1,12,16]*

1 Department of Biomedical and Clinical Sciences, University of Milan, Milan, Italy, 2 Department of Women's and Children's Health, University of Padua, Padua, Italy, 3 Department of Health Sciences, University of Milan, Milan, Italy, 4 Department of Biomedical, Surgical and Dental Sciences, Università degli Studi di Milano, Milan, Italy, 5 UOC Maxillo-facial Surgery and Dentistry, Fondazione IRCCS Cà Granda, Ospedale Maggiore Policlinico, Milan, Italy, 6 Department of Philosophy "P. Martinetti", Università degli Studi di Milano, Milan, Italy, 7 ASST GOM Niguarda, "Claudio Munari" Epilepsy Surgery Center, Milan, Italy, 8 Department of Medicine and Surgery, Unit of Neuroscience, University of Parma, Parma, Italy, 9 Neuroimmunology Group, Pediatric Research Institute "Città della Speranza", Padua, Italy, 10 Department of Neuroscience, University of Padua, Padua, Italy, 11 School of Philosophical, Historical and Indigenous Studies Monash University, Melbourne, Australia, 12 Canadian Institute for Advanced Research (CIFAR), Brain, Mind, and Consciousness Program, Toronto, Ontario, Canada, 13 Monash Centre for Consciousness and Contemplative Studies (M3CS), Monash University, Melbourne, Australia, 14 Sussex Centre for Consciousness Science, University of Sussex, Brighton, United Kingdom, 15 School of Engineering and Informatics, University of Sussex, Brighton, United Kingdom, 16 IRCCS Fondazione Don Carlo Gnocchi, Milan, Italy

⊕ These authors contributed equally to this work.
* marcello.massimini@unimi.it

## Abstract

Hemispherotomy is a neurosurgical procedure for treating refractory epilepsy, which entails disconnecting a significant portion of the cortex, potentially encompassing an entire hemisphere, from its cortical and subcortical connections. While this intervention prevents the spread of seizures, it raises important questions. Given the complete isolation from sensory-motor pathways, it remains unclear whether the disconnected cortex retains any form of inaccessible awareness. More broadly, the activity patterns that large portions of the deafferented cortex can sustain in awake humans remain poorly understood. We address these questions by exploring for the first time the electroencephalographic (EEG) state of the isolated cortex during wakefulness before and after surgery in 10 pediatric patients, focusing on non-epileptic background activity. Post-surgery, the isolated cortex exhibited prominent slow oscillations (<2 Hz) and a steeper broad-band spectral decay, reflecting a redistribution of power toward lower frequencies. This broad-band EEG slowing resulted in a marked decrease of the spectral exponent, a validated consciousness marker, reaching values characteristic of deep anesthesia and the vegetative state. When compared with

**Data availability statement:** The original EEG and imaging data are sensitive, as they pertain to pediatric patients. These data can only be made available on a public repository in aggregated form (https://doi.org/10.5281/zeno-do.16936516). The uploaded Excel files list the values of the original individual datapoints for all relevant figures in the main and supplementary text. The Matlab code and an equivalent Python code translation are available online on the same repository (https://doi.org/10.5281/zenodo.16936516).

**Funding:** M.M. is supported by the European Research Council (ERC-2022-SYG – 101071900 – NEMESIS; https://cordis.europa.eu/project/id/101071900). M.M. and E.M. are supported by the Ministero dell'Istruzione, dell'Università e della Ricerca PNRR – EBRAINS-Italy (IR00011; https://ebrains-italy.eu/resources/research-groups/43). Si.Sa. is supported by the Ministero dell'Istruzione, dell'Università e della Ricerca (PRIN 2022; https://prin.mur.gov.it/Iniziative/Detail?key=xntlwCEQJ%2BTLxeoukrh%2FdQ%3D%3D). A.K.S. is supported by the European Research Council (CONSCIOUS – 101019254; https://cordis.europa.eu/project/id/101019254/results). T.B., A.K.S, and M.M. acknowledge support provided by the Canadian Institute for Advanced Research (Brain, Mind, and Consciousness Program; https://cifar.ca/research-programs/brain-mind-consciousness/). A.P. is supported by HORIZON EUROPE European Research Council (HORIZON-INFRA-2022 SERV (Grant No.101147319) "EBRAINS 2.0: A Research Infrastructure to Advance Neuroscience and Brain Health"; https://cordis.europa.eu/project/id/101147319/it) and by the Ministero dell'Istruzione, dell'Università e della Ricerca (PRIN P2022FMK77; https://unifind.unito.it/resource/project/GARF_PNRR_PRIN22_23_01). The funders had no role in study design, data collection and analysis, decision to publish, or preparation of the manuscript.

**Competing interests:** I have read the journal's policy and the authors of this manuscript have the following competing interests: M.M. is co-founder and shareholder of Intrinsic Powers, Inc., a spin-off of the University of Milan; Si.Sa. is advisor of the same company. A.K.S. is an advisor to Conscium Ltd and AllJoined Inc.

a reference pediatric sample across the sleep–wake cycle, the spectral exponent of the contralateral cortex aligned with wakefulness, whereas that of the isolated cortex was consistent with deep NREM sleep. The findings of prominent slow oscillations and broad-band slowing provisionally support inferences of absent or reduced awareness in the isolated cortex. Moreover, the persistence of unihemispheric sleep-like patterns years after surgery provides unique insights into the long-term electrophysiological effects of cortical disconnections in the human brain.

## Introduction

Hemispherotomy is a surgical procedure used to treat severe cases of refractory epilepsy of structural etiology in children [1,2]. The goal of this procedure is to achieve maximal disconnection of the pathological cortex from the rest of the brain by severing its white matter connections with brainstem, basal ganglia, thalamus, and contralateral hemisphere. Typically, the extent of the cortical disconnection in hemispherotomy ranges from one lobe to an entire hemisphere [2,3]. The disconnected cortex is not surgically removed, as in hemispherectomy [4,5], but is left inside the cranial cavity with preserved vascular supply [6], completely devoid of communication with the external environment—except for cortical afferents from the olfactory bulb, in some cases. Hemispherotomy, currently preferred over hemispherectomy due to fewer clinical complications [7], is effective in blocking the spread of epileptic seizures from the pathological cortex to the rest of the brain, allowing patients to lead satisfying lives [2,8,9].

The intact contralateral cortex (which remains connected to the body and the external world) continues to support wakeful consciousness after surgery, as patients recover (at least) partial sensory-motor functions, cognitive abilities, and the capacity to report their internal states [2,10]. However, the disconnected cortex, isolated from sensory and motor pathways, cannot be evaluated behaviorally, leaving open the question of whether it retains internal states consistent with some form of awareness [11,12]. More broadly, the activity patterns that extensive regions of the human brain can sustain following massive deafferentation remain largely unknown.

The available evidence is drawn from preliminary fMRI studies. A first study, involving two pediatric participants, revealed preserved lateralized connectivity in resting state networks within the disconnected hemisphere, despite decreased regional cerebral blood flow [13]. A classifier aligned the pattern in the disconnected cortex with that of Minimally Conscious State (MCS) patients, indicative of a state of diminished consciousness. A subsequent study in a larger cohort found preserved functional connectivity in all resting-state networks, including the default mode network (DMN), in the disconnected hemisphere, albeit with reduced between-network segregation [14]. This finding was interpreted as indicative of a potential "island of awareness"—that is, the capacity for internally generated experiences (e.g., dreams) arising within an isolated cortical region [11]. However, inferring the internal state of the isolated cortex solely based on fMRI activity remains challenging. Indeed, functional

The remaining co-authors have no conflicts of interest to declare.

**Abbreviations:** aperiodic PSD component, the portion of the PSD with a 1/f-like shape; ANOVA, Analysis of Variance; Contra, contralateral cortex, homologous of the disconnected; Discon, Disconnected cortex; DMN, Default Mode Network; d.f., Degrees of Freedom for statistical tests; EEG, Electroencephalography; EMG, Electromyography; EOG, Electro-oculography; fMRI, Functional Magnetic Resonance Imaging; Hemispherotomy, the disconnection of a large portion of the cortex, up to an entire hemisphere; "island of awareness", a cortical area disconnected from the environment yet capable of awareness; ICA, Independent Component Analysis; IIR, Infinite Impulse Response, type of filter; MCS, Minimally Conscious State, diagnosis in the spectrum of Disorders of Consciousness; MRI, Magnetic Resonance Imaging; N1, Stage 1 Non-Rapid Eye Movement Sleep; N2, Stage 2 Non-Rapid Eye Movement Sleep; N3, Stage 3 Non-Rapid Eye Movement Sleep; NREM, Non-Rapid Eye Movement; periodic PSD component, the portion of the PSD exceeding the 1/f-like decay; Post, Post-hemispherotomy surgery; Pre, Pre-hemispherotomy surgery; PSD, Power Spectral Density, estimates of the power density across the frequency spectrum; Slow Delta Power, Mean PSD between 0.5–2 Hz; Spectral Exponent, reflects the decay of the aperiodic PSD component, i.e., the degree of broad-band slowing; Spindle grapho-elements, Rhythmic bursts of EEG (~10–16 Hz in pediatric population) during NREM sleep; TMS, Transcranial Magnetic Stimulation; 1/f, One over f, the aperiodic PSD component is inversely proportional to the frequency (f).

connectivity within resting-state networks is observed not only during wakefulness but can be retained also in states such as coma, general anesthesia, and deep Non-Rapid Eye Movement (NREM) sleep [15–18].

One way to clarify this complex landscape would be to assess the background electrophysiological states of the brain of awake subjects both before and after hemispherotomy. Beyond older studies on the background electroencephalographic (EEG) of the remaining contralateral hemisphere after hemispherectomy [19–22] and of partially disconnected prefrontal lobe during lobotomy [23,24], the literature on EEG activity in patients with hemispherotomy focused primarily on interictal epileptiform activity and clinical outcome [25–31], aiming at patient selection and prognostic assessment [4,32–34]. To date, the only studies examining lateralized changes in regional EEG activity in hemispherotomy patients were performed under general anesthesia, which precludes reliable inferences about the state of consciousness [35,36].

Here, we fill this gap by analyzing the EEG background—i.e., free of epileptiform discharges—recorded during wakefulness in 10 pediatric participants who had undergone a complete cortical disconnection, of either an entire hemisphere (in 7 cases) or a large portion of it (temporo-parieto-occipital disconnection in 3 cases).

Specifically, we investigated whether hemispherotomy led to the emergence of slow wave activity and to an overall broad-band slowing in the isolated cortex. The rationale for this exploration rests on two fundamental considerations drawn from the literature on experimental models and clinical populations. First, converging evidence suggests that cortical deafferentation can lead to resting electrophysiological patterns dominated by prominent slow oscillations (around 1 Hz), resembling those typically observed during deep sleep. Local intrusions of sleep-like slow waves during wakefulness have been documented in the cortex following focal brain injuries that disrupt both ascending and lateral connections [37]. This phenomenon has been observed in both animal models [38] and human patients [39]. Additionally, pronounced sleep-like slow oscillations represent the dominant activity pattern in experimental models of complete cortical disconnection [40], such as isolated cortical gyri in animal models (i.e., cortical slabs) [41] and cortical slices in vitro [42].

Second, evaluating slow waves in the background EEG is a widely used and practical approach for assessing the level of consciousness in clinical settings. While the presence of slow waves may not necessarily indicate a complete loss of consciousness [43] and conscious experiences can occur in deep sedation and deep sleep [44], a background EEG dominated by slow wave activity is broadly recognized as a key factor in clinically and ethically significant decision-making. This includes critical applications such as the stratification of patients with disorders of consciousness [45–47] and the adjustment of anesthetic depth during surgery [48,49]. Moreover, specific EEG slow wave patterns have been linked to the absence of reported conscious experience in within-state paradigms during NREM sleep [50].

To characterize background EEG activity before and after surgery, we employed not only a classical narrow-band spectral analysis but also the spectral exponent, which reflects the broad-band 1/f-like decay of the Power Spectral Density (PSD).

This quantitative measure offers several advantages in our context. First, by indexing the relative distribution of power across low and high frequencies, the spectral exponent is less influenced by potential reductions in cortical volume that could affect absolute EEG low-frequency power [48]. Second, research on serial awakenings during NREM sleep has shown that the likelihood of dream reports can be inferred from the balance between low- and high-frequency power [51].

Most importantly, the spectral exponent has been validated as a reliable EEG marker of the level of consciousness across a range of physiological [52–60], pharmacological [55, 61–65], and pathological states [62,64,66,67], including conditions of disconnected consciousness such as ketamine-induced dreaming and locked-in syndrome. Crucially, the spectral exponent has also been extensively studied in large pediatric samples, both during wakefulness [68–72] and NREM sleep [56,68].

In this work, leveraging quantitative EEG analysis and established reference values in a pediatric population, we investigated where the electrophysiological state of the isolated hemisphere falls along the spectrum between states associated with vivid awareness and those characterized by absent or reduced consciousness.

## Materials and methods

### Patients receiving disconnective surgery

We retrospectively selected our participants starting from a large sample of pediatric patients referred to the Munari Center for Epilepsy Surgery of Niguarda Hospital in Milan between 2005 and 2020. The recordings from 10 patients were selected on the basis of the following inclusion criteria: diagnosis of focal drug-resistant epilepsy with structural etiology on a malformative basis or as an outcome of perinatal hypoxic-ischemic insult; patient undergoing complete cortical disconnection of an hemisphere or of a large portion of it (at least one lobe); post-intervention outcome classified in class I according to the Engel Surgical Outcome Scale; availability of a wakefulness EEG within 12 months prior to surgery and at least 6 months after surgery; age at pre-surgery EEG recording between 2 and 17 years; presence of at least 3 min of quiet wakefulness free from artifacts and interictal epileptiform discharges.

During the EEG the participants, lying awake on a bed or in the arms of the parent, did not perform any particular task, except alternating periods of dozens of seconds with eyes open and with eyes closed, following a constant protocol across participants. Since we were not interested in the neural basis of specific contents of consciousness, we considered these epochs jointly (as in [68]). We excluded from the analysis all epochs in which the clinicians performed activation tests (intermittent light stimulation and hyperpnea). When necessary, younger children watched cartoons in order to improve compliance and reduce motion artifacts. Standard wakefulness EEG recordings, performed for routine clinical purposes, were used for research purposes after acquiring written informed consent from the participants' parents. The study was conducted in accordance with the Declaration of Helsinki and the institutional guidelines and the protocol was approved by the Ethics Committee of Niguarda Hospital, Milan, Italy (protocol number ID 348–24.06.2020).

### Data acquisition and pre-processing

Each EEG recording was performed in wakefulness and gathered from 19 scalp electrodes (cup in chlorinated silver) positioned according to the international 10–20 system and a 32-channel amplifier (Neurofax EEG-1200, Nihon Kohden Corporation). During acquisition, an additional electrode located between the Fp1 and Fp2 electrodes was used as a reference; the impedances of all the electrodes were kept below 5 kΩ for the entire duration of the recording. The signal was filtered during acquisition (high-pass filter 0.016 Hz; low-pass filter 300 Hz) and sampled at 500 Hz. The raw data was imported and analyzed with custom MATLAB code (MATLAB 9.7.0 R2019b, The MathWorks, Natick, MA, USA). The signal was filtered with a 5th-order Butterworth high-pass filter at 0.1 Hz (effectively retaining Slow Delta frequency activity between 0.5 and 2 Hz), with a 3rd-order Butterworth low-pass filter at 60 Hz and with a 50 Hz Notch filter. Epochs characterized by artifacts and interictal epileptiform discharges (spikes, polyspikes, sharp waves, spike/polyspike-wave complexes) were visually selected and excluded by a trained pediatric neurologist expert in EEG (Section A and Fig A in

). Due to the careful application of electrodes in our low-density set-up, recordings from all electrodes were deemed of sufficiently good quality. Electrodes were first re-referenced to the common average, to ease the topographical inspection of the components from a subsequent blind source separation. Specifically, to minimize the influence of electromyographic (EMG) and electro-oculographic (EOG) activity, Independent Component Analysis was performed on a copy of the data, filtered with a narrower high-pass filter (5th-order Butterworth at 0.5 Hz) to better separate artifactual from neurophysiological components. The different components of the signal were visually inspected on the basis of their time series, topographies, and PSD. Clearly identifiable components resembling EMG and EOG activity were marked for rejection. The unmixing weights (estimated from the data filtered with a 0.5 Hz high-pass) were applied to the original data (i.e., that filtered with a 0.1 Hz high-pass), and the retained components were back-projected to the scalp, thus effectively removing artifactual influences from the scalp electrodes signal [73].

Finally, to observe lateralized effects with high spatial specificity under our low-density EEG montage, bipolar derivations between all pairs of nearest neighboring electrodes (excluding the midline, Left hemisphere: {'Fp1-F3' 'F3-C3' 'C3-P3' 'P3-O1' 'F7-T3' 'T3-T5' 'F7-F3' 'T3-C3' 'T5-P3' 'Fp1-F7' 'F7-C3' 'F3-T3' 'T3-P3' 'C3-T5' 'T5-O1'}, Right hemisphere: {'Fp2-F4' 'F4-C4' 'C4-P4' 'P4-O2' 'F8-T4' 'T4-T6' 'F8-F4' 'T4-C4' 'T6-P4' 'Fp2-F8' 'F8-C4' 'F4-T4' 'T4-P4' 'C4-T6' 'T6-O2'}) were considered for subsequent analyses. The common average reference was used exclusively for ICA preprocessing and for topographic visual inspection and display (Fig 1B, center).

## Estimation of Slow Delta Power and Spectral Exponent

We estimated narrow-band EEG slowing by means of Slow Delta Power and broad-band EEG slowing by means of the Spectral Exponent. Both metrics were derived by the PSD, which was estimated using Welch's method, with a 3-s Hanning window and a 50% overlap, following linear detrending of the time-series in each periodogram. Only periodograms that were entirely free from artifacts and from epileptiform activity were included. The estimation of low-frequency activity (above 0.5 Hz) was free from artifacts induced by the filter (0.1 Hz high-pass, as previously mentioned).

Since slow wave activity has a frequency of ~1 Hz both in sleep [74–79] and in general anesthesia [48,49] and is described as EEG oscillations with a period between 0.5 and 2 s by the AASM manual for sleep scoring [80], we accordingly estimated Slow Delta Power as the mean absolute PSD between 0.5 and 2 Hz, subsequently transformed by log10 (results for absolute total power are shown in Section B and Fig B in S1 Appendix).

The Spectral Exponent, on the other hand, indexes the slope of the overall PSD decay across frequencies, and thus characterizes the broad-band relative distribution between high and low frequencies (1/f-like). The neurophysiological 1/f-like distribution originates from aperiodic and quasi-periodic activity [81,82].

The exponent was estimated from a linear fit of the PSD under double logarithmic axes, after discarding frequency bins with peaks corresponding to periodic activity, i.e., bins with large positive residuals from a preliminary linear fit, and their contiguous bins with positive residuals (Section C and Fig C in S1 Appendix; see [61]). The Matlab code for PSD fitting and an equivalent Python code translation are available online https://github.com/milecombo/spectralExponent/blob/master/README.md; https://doi.org/10.5281/zenodo.16936516

Here, we estimated the Spectral Exponent over the 0.5–20 Hz range, so as to include the Slow Delta range, while excluding higher frequencies which are prone to muscular contamination in children (we showed in Fig D in S1 Appendix, for completeness, a group-level broad-band PSD estimated over the 0.5–40 Hz range). A similar fitting range, 1–20 Hz, previously used in [68], was also investigated in our patients' sample for completeness (Section C and Fig E in S1 Appendix).

Subsequently, the median value of each feature (Slow Delta Power and Spectral Exponent) was taken across the lateralized bipolar derivations placed over the isolated cortex (15 derivations in the case of a hemispherotomy and 6 for a temporo-parieto-occipital disconnection) and the corresponding homologous bipolar derivations placed over the contralateral cortex.

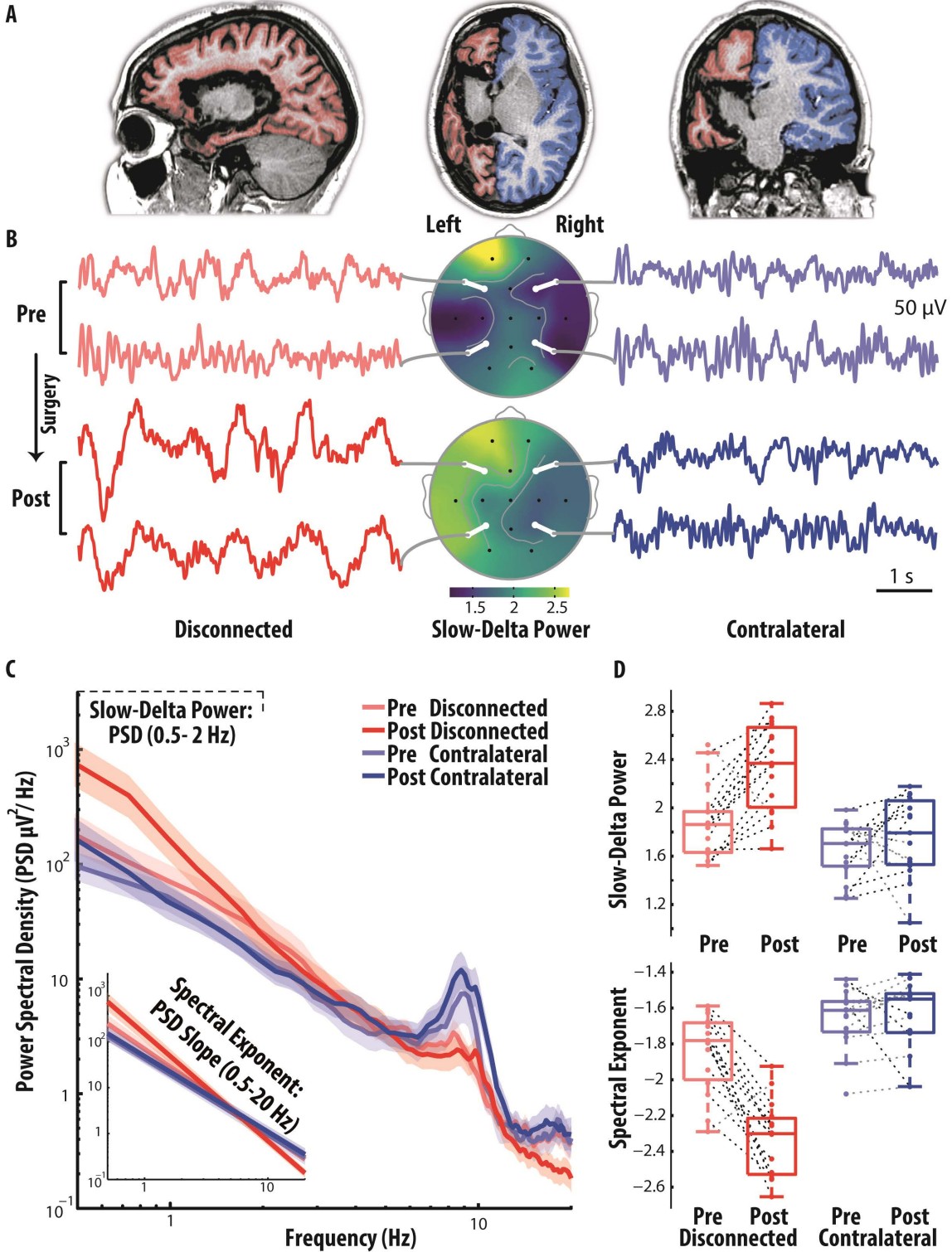

**Fig 1. Narrow- and broad-band spectral changes after hemispherotomy in an awake representative patient, displaying marked electroen-cephalography (EEG) slowing of the isolated cortex.** Cortical isolation of the left hemisphere in a 11 y.o. representative patient during wakefulness. **(A)** The anatomical MRI 6 months after surgery shows that all interhemispheric and subcortical connections were resected (sagittal, axial, and coronal

panels are shown from left to right). The cortical surface is highlighted in red for the disconnected cortex (in blue for the contralateral). **(B)** EEG recordings were performed during wakefulness, simultaneously for both hemispheres, 8 months before surgery (at 11 y.o.) and 12 months after. Short EEG segments (6 s) from the isolated cortex (F7-F3 and T5-P3 bipolar derivations, left panel) and the contralateral homologue (F8-F4 and T6-P4, right panel) are shown, pre-surgery (above) and post-surgery (below), selected from the recordings (respectively, 4 and 7 min after the exclusion of artifactual and epileptiform segments). In the middle panel, the topographies of Slow Delta Power, indexing the power spectral density (PSD) of low-frequency activity (0.5–2 Hz, hereby estimated under average reference for display purposes), show a marked inter-hemispheric asymmetry post-surgery. **(C)** The EEG PSD, averaged by geometric mean across intrahemispheric bipolar derivations, reveal an increase of slow frequency activity (<2 Hz) and a broad-band steepening of the PSD over the disconnected hemisphere post-surgery, which is evidenced in the inset graph by the linear fit of the PSD under double logarithmic axes (power-law fit). The shaded area indicates the bootstrapped 95% confidence interval for the geometric mean across electrodes. **(D)** For all the neighboring intrahemispheric bipolar derivations, we display the values of Slow Delta Power (0.5–2 Hz), and of the Spectral Exponent, indexing the slope of the PSD decay (0.5–20 Hz). The data underlying this figure are provided in the file Fig 1 Data, publicly available in the repository at the following https://doi.org/10.5281/zenodo.16936516.

## Statistical comparisons: Spatial inter-hemispheric differences, and temporal pre-post differences

We estimated narrow- and broad-band EEG slowing, by means of Slow Delta Power and of the Spectral Exponent, before and after surgical disconnection, in both the disconnected and the homologous contralateral cortex. We hypothesized that the disconnected cortex would display a larger degree of narrow- and broad-band slowing with respect to the contralateral cortex, and to its state before surgery, leading to larger spatial asymmetry after surgery. For each of the two EEG features, we thereby assessed whether EEG slowing differed between homologous cortices (spatial effect), between the recording sessions (pre- versus post-surgery, temporal effect), and if it differently changed in the two cortices following surgery (spatio-temporal interaction effect), by means of an ANOVA on a mixed effects model (Section D in S1 Appendix).

Then, according to the ANOVA results, a set of contrasts across spatial and temporal conditions was performed by means of planned paired *t* tests, for each EEG feature.

Specifically, we assessed a spatial inter-hemispheric contrast for each session (i.e., contrast of values between cortices, in the pre- and post-surgery session; Fig 2B and 2D), and a pre-post-surgery temporal contrast for each cortex (i.e., contrast of the pre-post difference against 0, in the disconnected and in the contralateral cortex, Fig 2C and 2E). Further, we ascertained whether the pre-post difference was larger for the disconnected cortex (i.e., contrast of the pre-post difference between cortices; Fig 2C and 2E); a finding which, in our specific design where measurements were paired over both time and space, would imply that the inter-hemispheric difference changed following surgical disconnection. We further compared the strength of statistical evidence between narrow- and broad-band spectral features by means of bootstrap analysis (Section E and Fig F in S1 Appendix). Moreover, we evaluated whether etiology and type of hemispherotomy had any influence on the main findings (Section F and Fig G in S1 Appendix).

## Spectral Exponent: Comparison to reference values during wakefulness and NREM sleep stages

To further establish whether Spectral Exponent values observed in patients with cortical disconnection were compatible with wakefulness or sleep, we included a reference dataset of Spectral Exponent values obtained from the EEG of a large sample of neurotypical pediatric participants during wakefulness and sleep [68]. In this previous study, the EEG was obtained during a daytime nap opportunity in an outpatient clinic, in children and adolescents aged from 2 to 17 years, who showed no neurological abnormalities. Here, to build the Spectral Exponent reference dataset, we considered only the subset of participants who were able to fall asleep and reached all NREM sleep stages (up to N3), consisting of 44 participants (24 males, aged between 2–16 years, mean: 6.798, SD: 3.944).

To directly compare Spectral Exponent values across datasets, we aligned the preprocessing pipeline of the patients with cortical disconnection to that of the previously published study on the reference pediatric sample [68]. Hence, in the patients' dataset, we increased the Butterworth high-pass filter from 0.1 to 0.5 Hz, and consequently restricted the fitting range of the Spectral Exponent from 0.5–20 Hz to 1–20 Hz (i.e., with the lower bound set at a frequency where the filter

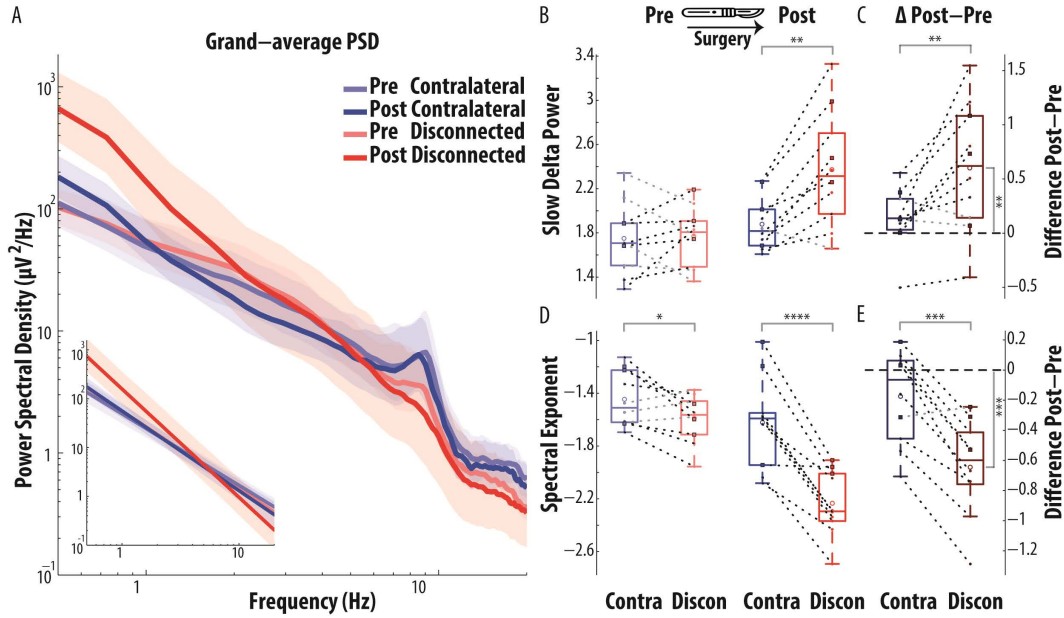

**Fig 2. Group-level narrow- and broad-band spectral changes after hemispherotomy reveal a marked EEG slowing of the isolated cortex, robust across patients. (A)** The geometric mean of the PSD was taken first across electrodes, and subsequently across patients, for the observed PSD (main graph) as well as for the aperiodic fit of the PSD (corresponding to a straight line under logarithmic scale for both *x* and *y* axes, inset graph). The shaded area indicates the bootstrapped 95% confidence interval for the geometric mean across participants. **(B)** The PSD in the Slow Delta band (0.5–2 Hz) was significantly larger in the disconnected (Discon) than in the contralateral (Contra) cortex, only in the session after surgery. **(C)** Following surgery, slow wave activity increased in the disconnected cortex, as indexed by a significant pre to post increase of the Slow Delta PSD in the disconnected cortex (i.e., positive post-pre differences in the disconnected cortex). This increase was larger in the disconnected than in the contralateral cortex. **(D)** The Spectral Exponent (0.5–20 Hz) was significantly more negative in the disconnected than in the contralateral cortex, in the session before surgery (Pre) and, particularly, in the session after surgery (Post), whereby consistent inter-hemispheric asymmetry was observed in all patients. **(E)** Following surgery, the PSD became steeper in the disconnected cortex, as indexed by a significant pre- to post-decrease in Spectral Exponent toward more negative values, observed in all patients (i.e., all negative post-pre differences for the disconnected cortex). This decrease was larger in the disconnected than in the contralateral cortex. Patients who received a temporo-parieto-occipital disconnection (squares with a black outline) displayed an overall similar pattern to that of patients who received a hemispherotomy (small dots). Significant contrasts are denoted by star symbols (* *P* < 0.05; ** *P* < 0.01; *** *P* < 0.001; **** *P* < 0.0001) over a bracket connecting the contrasted values, either the two cortices, or the post-pre difference against zero. The data underlying this figure are provided in the file Fig2_Data, publicly available in the repository at the following https://doi.org/10.5281/zenodo.16936516.

effects were dissipated and the expected PSD decay was observed), according to the procedure used in the reference dataset [68]. Even after restricting the frequency range under study, the findings on the Spectral Exponent were consistently replicated in the patient dataset (Section C and Fig E in S1 Appendix).

Furthermore, we digitally re-referenced the EEG recordings from the pediatric reference dataset to match the bipolar montage used in the patient dataset, ensuring methodological consistency for comparative analyses (see section Data acquisition and pre-processing). Hence, the Spectral Exponent of the reference dataset (previously estimated using average reference) was re-estimated over all lateralized nearest-neighbor bipolar derivations (15 per side, 30 total) and the median value was considered.

To quantify the proximity of the disconnected and contralateral cortex to reference physiological values of wakefulness and sleep, we estimated the differences of the means of each condition between the two datasets across bootstrap resamplings (Section G and Fig H in S1 Appendix).

Finally, given the well-known impact of age on EEG features [68,83,84], we conducted age-matched supplementary analyses to factor out a potential effect of age in our findings (Section H, I and Figs I, J, and K in S1 Appendix).

## Further comparison of the disconnected hemisphere with physiological NREM sleep

In the previous part of the study, we compared the Spectral Exponent in behaviorally awake patients before and after surgical disconnection to that of a reference pediatric cohort across wakefulness and NREM sleep stages. After observing that in awake patients the broad-band Spectral Exponent of the disconnected cortex closely resembled that of deep NREM sleep (of N2 and N3 stages particularly, Fig 3), we then explored how the EEG activity of the disconnected cortex differed from physiological sleep, considering NREM sleep stages N2 and N3 together.

This allowed us to ascertain the specific effect of deafferentation of the cortex from subcortical structures, particularly the thalamus, in the formation of two EEG hallmarks of physiological sleep, namely slow wave activity and spindles. We thus investigated slow wave activity and spindles, further characterizing the underlying spectral and temporal features of the Slow Delta band (in the main text) and the sigma band (in the S1 Appendix). We extracted these features in the disconnected cortex and compared them to those of wakefulness and NREM sleep of the pediatric reference dataset.

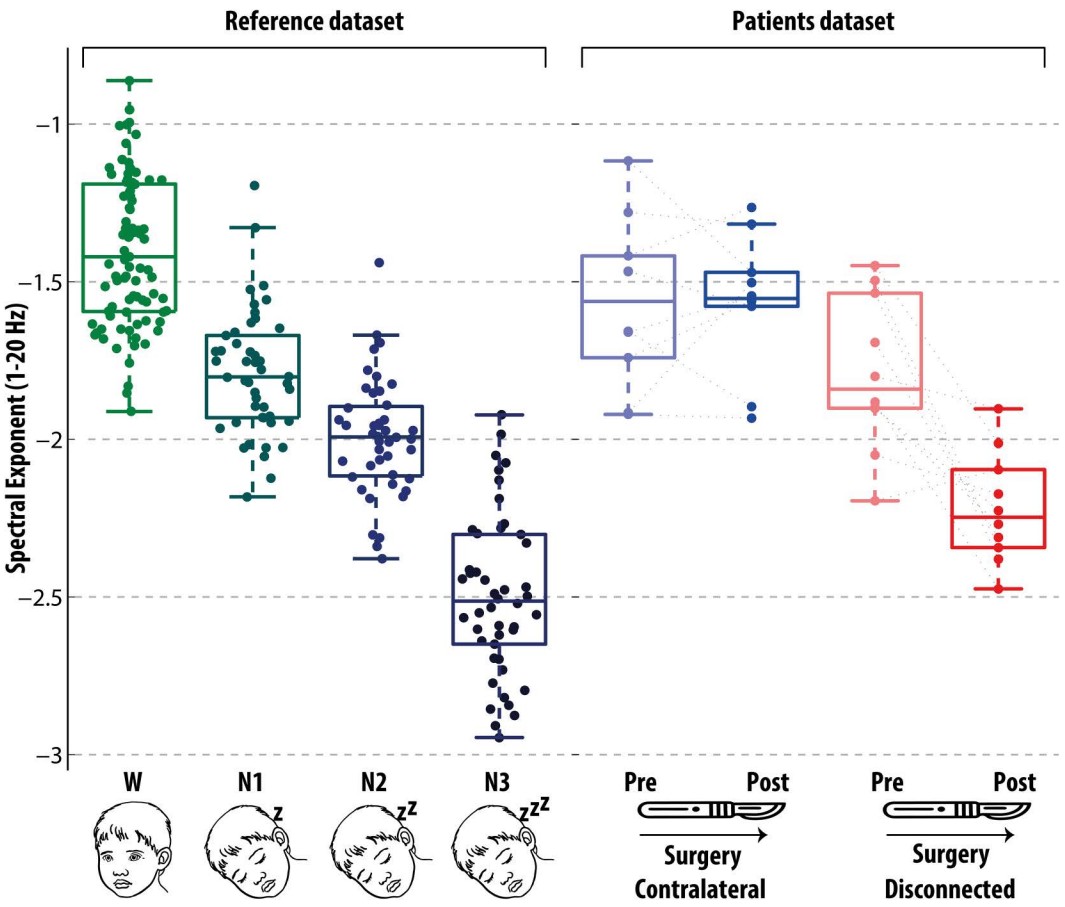

**Fig 3. The comparison of the Spectral Exponent in hemispherotomy with a reference pediatric sample across wakefulness and NREM Sleep reveal values compatible with deep sleep in the isolated cortex.** The Spectral Exponent is estimated in patients before and after surgery (Patients dataset, right panel), and in a reference pediatric sample across wakefulness and NREM sleep (Reference dataset, left panel). To enable comparison, the Spectral Exponent is specifically estimated over the 1–20 Hz band, and the cutoff frequency of the pre-processing high-pass filter has been raised to 0.5 Hz, in keeping with [68]. The contralateral cortex, both before and after surgery, showed values compatible with wakefulness. Before surgery, the pathological cortex showed values intermediate between those of wakefulness and those of N1 sleep. After surgery, the disconnected cortex showed values intermediate between those of N2 and N3 sleep. The data underlying this figure are provided in the file Fig 3 Data, publicly available in the repository at the following https://doi.org/10.5281/zenodo.16936516.

### Slow Delta oscillations and slow waves

**Amplitude of Slow Delta Activity.** To compare the strength of Slow Delta Activity in the disconnected cortex with that of physiological sleep and wakefulness, we estimated the amplitude of the envelope of Slow Delta oscillations. For this purpose , we first low-pass filtered the EEG signal (IIR forward and reverse Butterworth filter, 5th order) with a cutoff at 2 Hz, in alignment with the spectral domain analysis (Figs 1 and 2), then considered the absolute value of the Hilbert transform. A density plot of the typical span of the amplitude values across subjects is shown, for both the patient and reference dataset (Section J and Fig M in S1 Appendix). The mean Slow Delta amplitude was obtained by averaging the amplitude envelope over time, then its square root was taken, to reduce the skewness of the distribution among subjects; finally, the median value across channels was retained. Further, we corroborated this analysis by means of spectral analysis in the Slow Delta band (Section J in S1 Appendix).

**Period of Slow Delta oscillations.** After characterizing the amplitude of Slow Delta fluctuations, we then characterized the period of Slow Delta Activity, irrespective of amplitude, of the disconnected cortex with respect to physiological NREM sleep and wakefulness. To estimate the period of activity showing oscillations in the Slow Delta band, the signal was first low-pass filtered in the delta band (IIR forward and reverse Butterworth filter, 5th order, cutoff at 4 Hz), and then only oscillations with a period consistent with Slow Delta oscillations were considered (the time between the zero-crossings of a full oscillation was required to be longer than 0.5 s, corresponding to frequencies lower than 2 Hz, adapted from [85–87]). Different from mastoid-referenced EEG, where only half-period fluctuations within the negative polarity are considered [88–90], in our case of bipolar channels we considered full-period signal fluctuations across positive and negative polarity. We then estimated the mean period across all Slow Delta oscillations; a density plot of the typical span of the period values across subjects is shown, for both the patient and reference dataset (Section K and Fig N in S1 Appendix). To compare values between subjects of different conditions, the median value across channels was retained (Fig 4).

Finally, given the well-known impact of age not only on Spectral Exponent but also on Slow Delta oscillations [83,91,92], we conducted supplementary analyses to control the effects of age on our findings. These analyses corroborated the main results on slow wave properties of the disconnected cortex (Section L and Fig O in S1 Appendix). Moreover, given that in sleep the hotspot with maximal slow wave activity migrates across the developmental age from posterior to anterior regions in healthy children [83], we performed supplementary analyses considering a subset of electrodes in the anterior and posterior regions, in order to verify potential regional effects (Section M and Fig P, Q in S1 Appendix).

**Sigma oscillations and spindles activity.** Given the established thalamic origin of sleep spindles, we confirmed their absence in the disconnected cortex through multiple assessments: visual inspection by a trained neurologist, analysis of total and periodic power in the sigma band, and automated detection of sigma-band bursts (Section N in S1 Appendix).

## Results

### Patients and EEG recordings

Among the 10 patients fulfilling inclusion criteria, 4 had a diagnosis of focal drug-resistant epilepsy with structural etiology on a malformative basis (i.e., pachygyria, polymicrogyria, or focal cortical dysplasia) and 6 as an outcome of perinatal hypoxic-ischemic insult. Patients underwent a complete disconnection of a large cortical portion: either an entire hemisphere (7 patients), or the temporo-parieto-occipital region (3 patients). The EEG recordings were performed in awake patients within 12 months before surgery (mean = 5.6, SD = 4.67, range = [1, 12]) and at least 6 months after surgery (delay in months: mean = 19.8, SD = 9.6, range = [6.96, 34.92]). The age at pre-surgery EEG spanned from 2 to 11 years (mean = 7.8, SD = 2.9). In each recording, we identified at least 3 min of quiet wakefulness free from artifacts and interictal epileptiform discharges (mean = 7.379, SD = 3.489, range = [3.561, 12.815]) to focus on background EEG activity. A description and a visual illustration of the epileptiform activity, excluded from further analyses, is provided in Fig A and Section A in S1 Appendix). Demographic and clinical information for each patient are reported in Table A in S1 Appendix.

PLOS Biology

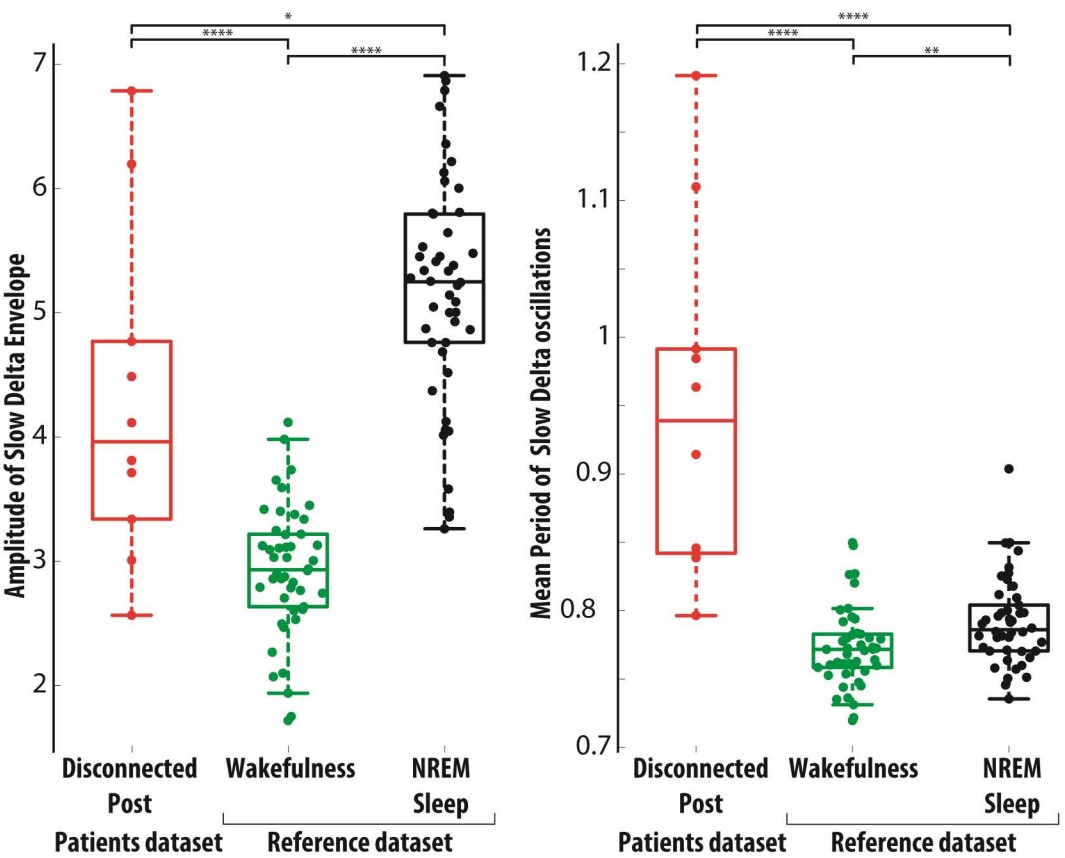

**Fig 4. Mean amplitude of the Slow Delta Envelope and period of Slow Delta oscillations in the disconnected hemisphere, compared with respect to wakefulness and NREM sleep in the reference pediatric sample. (A)** The mean amplitude of the envelope filtered in the Slow Delta band, median across channels, is shown for each patient in the disconnected cortex after surgery, and, for comparison, in the reference dataset during wake- fulness and NREM sleep. **(B)** The mean period of Slow Delta oscillations, median across channels, is shown for each subject across conditions. The median across channels is shown for each subject across conditions. See Figs M and N in S1 Appendix for additional details. The data underlying this figure are provided in the file Fig 4_Data, publicly available in the repository at the following https://doi.org/10.5281/zenodo.16936516.

## Surgical disconnection and EEG slowing

A representative example of the main results is shown for an 11-year-old girl with a left porencephalic lesion secondary to perinatal stroke undergoing hemispherotomy (Fig 1A). The visual inspection of the EEG time-series recorded in an awake patient and of the topographies of low-frequency PSD (Slow Delta Power, 0.5–2 Hz) reveal high-amplitude (>50 µV) slow waves emerging over the disconnected cortex post-surgery, associated with pronounced inter-hemispheric asymmetry (Fig 1B). Further, the PSD profiles (Fig 1C) indicate that the narrow band increase in the Slow Delta band was accom- panied by an overall redistribution of power from high to low frequencies, resulting in a steeper broad-band decay of the disconnected cortex following surgery. These observations were quantified by the Slow Delta Power and the Spectral Exponent (indexing the slope of the PSD decay over the 0.5–20 Hz range) over intra-hemispheric bipolar derivations (Fig 1D).

These single-subject findings were consistently observed at the whole sample level. When comparing the seven patients with hemispherotomy of an entire hemisphere to those who underwent temporo-parieto-occipital disconnection, we observed a highly similar pattern of localized increase in the degree of EEG slowing in the disconnected cortex (Section F in S1 Appendix). Results are thus reported across all 10 patients, irrespective of the extent of the surgery. As shown in Fig 2, at the population

level, in awake patients, the disconnected cortex following surgery displayed increased Slow Delta Power and an overall PSD redistribution toward low frequencies and a steeper broad-band decay, as evident from the PSD and its 1/f-like fit, aggregated across patients. These observations were confirmed by statistical analysis, reported in the next section.

## Asymmetry of cortical states after disconnection

For each of the two EEG dependent variables (Slow Delta Power and Spectral Exponent, indexing narrow- and broad-band slowing, respectively), the ANOVA on the mixed effects model revealed a main effect of time (both $p < 0.001$), demonstrating a change due to surgical disconnection. Further, it also revealed a significant interaction effect between time and space (both $p < 0.001$), indicating that surgery differently affected the disconnected versus the contralateral cortex, and, similarly, that the inter-hemispheric asymmetry changed following surgery (for details see section S4 in S1 Appendix).

The specific effects of cortical disconnection on each EEG feature were explored through a series of planned pairwise contrasts, $t$ tests with 9 d.f. (Fig 3 and Table C in S1 Appendix).

Contrasting the pre-post difference between cortices, the disconnected cortex underwent a greater change following surgery as compared to the contralateral, for both the Slow Delta Power ($T(9)= -3.146$, $p = 0.0118$, greater change observed in 8/10 patients, Fig 2C) and the Spectral Exponent ($T = 6.464$, $p = 0.0001$, greater change observed in 9/10 patients, Fig 2E), indicating also that the inter-hemispheric asymmetry increased following surgery.

When assessing spatial inter-hemispheric contrasts (contralateral versus disconnected cortex), we found that Slow Delta Power was significantly larger in the pathological cortex than in the contralateral cortex, but only in the post-surgery session (Pre: $T = -0.228$, $p = 0.8246$; larger in 6/10 patients; Post: $T = -4.245$, $p = 0.0022$, larger in 9/10 patients, Fig 2B); the Spectral Exponent was significantly more negative in the disconnected cortex than in the contralateral, both pre-surgery ($T = 2.395$, $p = 0.0402$; more negative in 7/10 patients) and, especially, post-surgery ($T = 7.118$, $p < 0.0001$, more negative in 10/10 patients, Fig 2D).

Assessing temporal contrasts (Post-Pre surgery), the Slow Delta Power was larger following surgery, significantly only in the disconnected cortex (contralateral: $T = -1.462$, $p = 0.1777$, larger in 9/10 patients; disconnected: $T = -3.215$, $p = 0.0106$; larger in 10/10 patients, Fig 2C); similarly, the Spectral Exponent was significantly more negative following surgery, only in the disconnected cortex (contralateral: $T = 1.786$, $p = 0.1077$, more negative in 5/10 patients; disconnected: $T = 6.455$, $p = 0.0001$, more negative in 10/10 patients, Fig 2E).

In supplementary control analysis, neither the etiology (cortical malformation, perinatal stroke) nor the type of hemispherotomy (whole-hemisphere, and temporo-parieto-occipital disconnection) appeared to influence the main results, as all subgroups qualitatively showed the same post-surgical asymmetry and pre–post changes in the disconnected cortex observed at the group level (Section F in S1 Appendix). We moreover verified that the extent of disconnected cortex did not play a significant role, by means of a mixed-effect model analysis including the type of hemispherotomy as a covariate (Section F in S1 Appendix).

Overall, before surgery, we found no evidence for interhemispheric difference in narrow-band Slow Delta Power and only a small, albeit significant, inter-hemispheric asymmetry in the degree of broad-band slowing. In contrast, after surgery, we observed a marked inter-hemispheric asymmetry characterized by prominent increase in both the degree of EEG narrow-band and, especially, of broad-band slowing over the disconnected cortex. The Spectral Exponent revealed larger effect sizes than Slow Delta Power, thus potentially providing a more sensitive index of the neurophysiological consequences of hemispherotomy (Section E and Fig F in S1 Appendix).

## The Spectral Exponent of the disconnected cortex in awake patients aligns with values of physiological NREM sleep

Finally, the Spectral Exponent values assessed in awake patients before and after cortical disconnection were compared to the values assessed in a reference pediatric sample during wakefulness and NREM sleep (Fig 3). Qualitatively, the typical values (i.e., those of the central half of the sample, within the interquartile range) of the contralateral cortex

overlapped with the typical values observed during wakefulness and N1 in the reference sample. The typical values of the disconnected cortex before surgery overlapped with the typical wakefulness and N1 values of the reference sample; after surgery, the typical values of the disconnected cortex overlapped with the typical N2 and N3 values of the reference sample. Notably, 9/10 values from the disconnected cortex did not overlap with the wakefulness distribution; and the 10th case was below all but one of the values from the wakefulness distribution. Age-matched analysis further strengthened this result, revealing that the disconnected cortex showed lower Spectral Exponent values in all 10 cases compared to their reference counterparts during both wakefulness and N1 (Section I and Fig K in S1 Appendix).

Subsequently, we compared the mean Spectral Exponent values of each condition across the two datasets through bootstrap resampling. Specifically, we considered the bootstrapped distribution of the pairwise differences between the mean values of the pediatric reference participants (either during wakefulness, N1, N2, or N3 sleep) and the mean values of the patients (in the disconnected or contralateral cortex, before or after surgery) (Section G and Fig H in S1 Appendix). According to bootstrapped differences of mean values, the homologous contralateral cortex, both before and after surgery, showed values distinct from the NREM sleep distributions and closest to the wakefulness distribution, albeit on the lower end. Moreover, the disconnected cortex before surgery showed values similar to the N1 sleep distribution; crucially, after surgery, the disconnected cortex showed values distinct from the wakefulness distribution and intermediate between those of the N2 and N3 sleep distributions.

Supplementary age-matching analysis confirmed these findings (Section I and Fig K in S1 Appendix); regardless of age, while the Spectral Exponent values in the disconnected cortex resembled those of N1 sleep before surgery, they shifted toward values characteristic of deeper NREM stages (N2–N3) after surgery.

## Slow Delta Activity in the disconnected cortex

We further characterized slow wave activity occurring during wakefulness in the disconnected cortex, by comparing the average amplitude of the envelope of EEG activity filtered in the Slow Delta band with respect to the pediatric reference sample across both wakefulness and sleep (Fig 4). Confirming the results of spectral domain analysis, large amplitude values occurred more frequently in the pathological cortex following surgical disconnection, as displayed in the grand-average density plot (Section J in S1 Appendix). Most notably, larger amplitude values were found with respect to the wakefulness of the pediatric reference sample ($T(54) = -5.3132$, $P = 2.0934e{-}06$). When directly compared to physiological N2 and N3 sleep, slow wave activity appearing during wakefulness in the disconnected cortex was on average smaller ($T(54)=2.5986$, $P = 0.012043$). Similar findings were obtained considering the PSD in the Slow Delta band (Section J in S1 Appendix).

Next, we characterized the typical period of Slow Delta oscillations by analyzing activity in a wider delta band and considered only Slow Delta oscillations (period >0.5 s). Large period values occurred more frequently in the pathological cortex following surgical disconnection, as displayed in the grand-average density plot (Section K in S1 Appendix). The mean period of Slow Delta oscillations of the disconnected cortex was markedly larger than that observed during either wakefulness ($T(54) = -8.566$, $P = 1.2167e{-}11$) or sleep ($T(54) = -7.5059$, $P = 6.2164e{-}10$).

Supplementary age-matched analysis confirmed the key findings on the properties of slow-delta activity (Section L and Fig O in S1 Appendix): regardless of age, in the disconnected cortex, amplitude values were intermediate between those observed during wakefulness and NREM sleep—with more pronounced differences relative to wakefulness—and the oscillation period was considerably longer than in either state.

Overall, the disconnected cortex in awake patients showed a remarkable emergence of Slow Delta Activity, leading to a state departing from wakefulness, yet different from full-fledged NREM sleep. Specifically, the disconnected cortex lacked spindle activity (Section N in S1 Appendix) and showed slow waves with reduced amplitude and longer period compared to NREM sleep, coherent with the absence of thalamic synchronizing input.

## Discussion

Following surgery, prominent slow waves appeared over the disconnected cortex, marked by a pronounced increase in Slow Delta Power (0.5–2 Hz). The contralateral cortex, retaining intact subcortical inputs, did not display any significant increase in the degree of EEG slowing following surgery, despite the surgical resection of all its inter-hemispheric connections. This finding, in line with neurophysiological evidence in callosotomy [93,94], suggests that the primary cause of EEG slowing observed in the disconnected cortex lies in the deafferentation from subcortical, rather than cortical, inputs.

During NREM sleep, EEG slow waves emerge throughout the cerebral cortex as a consequence of decreased levels of neuromodulation from brainstem activating systems [95]. A similar pattern can be observed in pathological conditions following lesions or compressions in the brainstem and midbrain ascending activating systems [96,97]. More generally, slow waves can be found in various experimental models in which cortical circuits are disconnected from subcortical inputs, such as in isolated cortical gyri (i.e., cortical slabs) [41], in cortical slices in vitro [42] as well as after thalamic inactivation [98]. As such, cortical sleep-like slow waves are considered the elemental intrinsic regime of cortical circuits [40]. The present study in hemispherotomy provides further evidence for the account of cortical slow waves as the default activity pattern of the cortical network [40]. Remarkably, this is, to our knowledge, the first evidence that this pattern can persist for months and years (recordings were performed between 6 and 36 months post-surgery) after complete cortical disconnection.

Below, we discuss the significance of these findings in relation to the ongoing debate about the potential for "islands of awareness" [11,12,99] following hemispherotomy and in the broader perspective of the consequences of structural brain disconnections.

### The problem of inaccessible awareness

Inferring consciousness in isolated human cortical hemispheres represents a unique challenge [11,12]. For example, the surgically disconnected cortex differs from lab-grown cortical organoids, because it has developed as part of a human brain in contact with the external world (and a physical body), and thus some arguments against the possibility of consciousness in cortical organoids would not apply to hemispherotomy [100].

A vast literature on acquired brain-injury [101–103], hemispherectomy [4,104–107] and split-brain patients [108,109] as well as studies conducted on subjects during the WADA test [110–112]) suggests that one hemisphere is sufficient to support consciousness. However, hemispherotomy also differs from split-brain surgery, where the two hemispheres are disconnected from each other but retain all subcortical afferents and their ability to interact with the body and the external world [108,109]. Hence, whether the isolated cortex represents a distinct locus of consciousness from that supported by the contralateral hemisphere remains an open question. Two recent fMRI studies indicate preserved neural networks lateralized within the isolated hemisphere and specifically the integrity of the DMN) [13,14]. While the first study [13] interprets this result as indicative of a lower level of consciousness, the second [14] contemplates the possibility of an island of awareness. A possible rationale for the second interpretation is that, in healthy awake subjects, the DMN is typically activated during self-directed mentation, such as daydreaming and reflection [113]. Yet, resting state networks, including the DMN, can be disrupted in psychedelic states in which vivid experiences are present [114] and, crucially, preserved in states where consciousness is lost, or diminished, such as sleep and general anesthesia [15–18] making it difficult to infer awareness solely based on fMRI patterns.

Assessing EEG activity patterns currently represent the primary tool to characterize physiological [80,115], pharmacological [116–119], and pathological [45–47,66,120] states of consciousness, both in the research and clinical settings. It is therefore important to consider how the electrophysiological state of the isolated cortex compares with those observed in these conditions. Here, we found pronounced broad-band EEG slowing, indexed by Spectral Exponent values similar to those found in conditions such as deep NREM sleep, general anesthesia, and the vegetative state, according to previous work [61, 63, 65–67].

Although NREM sleep, general anesthesia, and the unresponsive wakefulness syndrome do not always imply complete unconsciousness—since dream reports and signs of covert consciousness can be observed in a significant number of cases [44,51,121–126]—they constitute the principal counterfactual (apart from isoelectric brain death) to states, where consciousness is certain due to the presence of overt, immediate reports.

Of these states, deep NREM sleep represents the primary physiological instance in which a neurotypical human can authentically deny being conscious. Indeed, previous studies of serial awakenings during full nights of sleep result in a significant rate of dream reports (12%–25%, 45%, 34%, according to [50,125,127] respectively), reports of vague dreams without content (50%, 33%, 39%), but also in reports of no conscious experience (50%, 22%, 28%). A recent extensive survey revealed no conscious experience in ~70% during N3 sleep [128]. Crucially, recent studies employing a within-state paradigm during NREM sleep have identified EEG features that are highly predictive of unconsciousness.

Specifically, increased low-frequency power and decreased high-frequency power are predictive of the absence of dream reports during sleep [51]. Intracranial recordings in humans and animal models show that EEG slow waves reflect the occurrence of silent periods in cortical neurons (OFF-periods), which prevent cortical circuits from engaging in recurrent, complex patterns of network interactions [129–133]. Convergent empirical evidence and theoretical principles suggest that the preservation of these recurrent, complex dynamics is a necessary condition for consciousness [134,135] providing mechanistic support to the association between slow waves and the lack of dream reports.

Notably, in within-state studies during NREM sleep, the absence of dreams was specifically reported with an overall prominent low-frequency power—due to the presence of background cortico-cortical slow waves with slow rising time— not associated with an increase in high-frequency activity and spindles [50]; this pattern closely resembled that seen after hemispherotomy. Therefore, the present finding of a redistribution of power from high to low frequencies detected by the Spectral Exponent (Figs 1 and 2), in conjunction with suppressed spindle activity, can be interpreted as indicating absent or reduced likelihood of dream-like experiences in the isolated cortex.

## Properties of post-disconnection slow waves

The generation of sleep-like slow waves following structural injuries and disconnections in awake individuals is a long-standing finding [38,136]. In fact, the term delta waves—now commonly associated with the EEG patterns of deep NREM sleep—was originally introduced nearly a century ago by William Grey Walter to describe the slow waves recorded during wakefulness over focal brain lesions [136]. The idea that cortical sleep-like dynamics can intrude during wakefulness after brain injury, leading to network disruptions, loss of consciousness, or cognitive and motor deficits, has recently been systematically revisited [37]. Delta waves during wakefulness have been observed in various conditions, such as traumatic brain injury, diffuse axonal injury, and ischemic or hemorrhagic stroke [37,38,137–139]. However, their relationship to the key neurophysiological features of physiological sleep has yet to be systematically assessed.

The surgical procedure of hemispherotomy offers a unique model for investigating and characterizing the intrusion of sleep-like slow wave EEG activity during wakefulness following structural disconnection. In this study, we leveraged this model by directly comparing EEG recordings from awake hemispherotomy patients with wakefulness and NREM sleep recordings in a pediatric reference dataset, with an age compatible to that of the patients [68].

Following surgery, the Spectral Exponent of the isolated cortex deviated from the wakefulness distribution observed in the pediatric reference sample in 9 out of 10 cases (Fig 3, minimally overlapping in the remaining case). Age-matched analysis further supported this finding, revealing lower Spectral Exponent values in all 10 cases compared to their reference counterparts during both wakefulness and N1 sleep (Fig K in S1 Appendix). Conversely, the Spectral Exponent of the isolated cortex fell in the range consistent with N2-N3 sleep, as confirmed by bootstrap and by age-matched analysis (Fig H in S1 Appendix).

However, the sleep-like slow wave pattern recorded after disconnection also exhibited key differences compared to those recorded in healthy subjects. As expected under conditions of thalamic disconnection, post-surgery slow waves

were not accompanied by spindle activity (according to visual inspection, sigma burst detection, and spectral analysis of total and periodic power, Section N in S1 Appendix). Interestingly, although the isolated cortex exhibited a broadband spectral decay comparable to that seen in the sleeping reference condition, it displayed Slow Delta oscillations of lower amplitude and longer duration than those typically observed during natural NREM sleep (Fig 4). These findings under-score the essential role of thalamocortical circuits in generating and regulating EEG slow oscillations during NREM sleep. The structural integrity of cortico-thalamo-cortical loops appears to facilitate the synchronization of cortical activity, thereby increasing the amplitude of slow oscillations. In addition, thalamic neurons contribute to the temporal dynamics of these oscillations by initiating cortical up-states [140] and by extending down-states, as demonstrated by focal pharmacological inactivation of the thalamus [98,141]. The reduced amplitude and longer period of slow oscillations in the disconnected cortex may also reflect cortical dynamics with reduced rising time and propagation velocity. Animal studies suggest that cortical isolation or thalamic inactivation reduces neuronal synchrony and slows the propagation of slow waves [142]. Without long-range thalamic afferents, slow oscillations may arise from multiple asynchronous local cortical sources, leading to a spatiotemporal dispersion of slow waves. This low synchrony at the cortical surface level can contribute to the finding of EEG slow waves from the disconnected cortex having lower amplitude and longer duration than those from physiological sleep.

Beyond the effects of deafferentation from ascending thalamic input, diminished lateral excitatory connectivity may also contribute [98]. In cortical slabs, the period of slow oscillations has been found to scale inversely with the size of the isolated gray matter [41]. Indeed, small cortical slabs display long silent periods resembling a pattern of burst suppression, whereas continuous slow waves emerge if the size of the isolated tissue is increased.

Our discovery of slow waves persisting for years in the human isolated cortex raises significant questions. Specifi-cally, the functional significance of cortical slow waves intruding during wakefulness—in absence of the subcortical inputs required for physiological sleep regulation and spindle activity—remains unclear. Given that similar sleep-like dynam-ics are commonly observed following a wide range of brain injuries [37], it will be important to determine whether these patterns retain neuroplastic or homeostatic functions [143–147] or whether they merely represent a regression of cortical activity to its default "intrinsic" state [40,98,148]. Addressing this question is particularly important for understanding the role of slow waves after brain injury and their potential implications for rehabilitation, and will require focused longitudinal investigations.

Finally, the present findings prompt further investigation into the relationship between EEG slowing and fMRI func-tional connectivity patterns. While macroscale fMRI resting-state networks, including the DMN, remain preserved in the isolated hemisphere, intrahemispheric connectivity increases, and the normal anticorrelations between networks are lost [14]. Recent rodent studies suggest a link between functional network abnormalities, such as DMN hyperconnectivity, and EEG slow waves [149]. Similarly, stroke patients show EEG slowing and altered connectivity patterns, including the loss of normal anticorrelations, reduced interhemispheric connectivity, and increased intrahemispheric connectivity [37]. In this context, the unilateral sleep-like state observed after hemispherotomy offers a unique model for exploring the interplay between electrophysiological dynamics and fMRI functional networks across both physiological and pathological brain states.

## Limitations and future directions

Even before surgery, the pathological cortex exhibited slightly lower mean Spectral Exponent values compared to its contralateral counterpart, with a distribution centered between those of wakefulness and N1 sleep in the reference pedi-atric sample (Fig 3). This relative slowing under baseline conditions may be partially influenced by interictal epileptiform discharges, although these pathological grapho-elements were meticulously excluded during data pre-processing by a trained neurologist (see Fig A in S1 Appendix). Additional contributing factors could include focal cortical malformations and gliosis [150–152]. Alternatively, some degree of slowing may be contributed by local slow waves that play a protective

role against aberrant epileptiform activity [153]. In this context, the contribution of the pathology to the observed slowing before surgery, and its relevance to consciousness remain open questions.

Future investigations may benefit from a careful prospective design. Specifically, future prospective studies should include sleep recordings and repeated EEG assessment, to assess potential state-dependent changes and their evolution over time. Moreover, this study included only two disconnection types (hemispherotomy of an entire hemisphere and temporo-parieto-occipital disconnection) and did not consider the details of the surgical procedure; larger and more heterogeneous cohorts could clarify the roles of surgical technique, volume of the disconnected tissue and disconnection topography in shaping electrophysiological outcomes.

The main finding of this study concerns the relative changes observed in the disconnected cortex after surgery compared to the contralateral hemisphere.

As discussed above, the EEG evidence provided here is compatible with a state of absent or reduced awareness. However, any inference about the presence or absence of consciousness, based solely on the brain's physical properties (including prominent EEG slow waves), must be approached with caution, particularly in neural structures that are not behaviorally accessible. For example, the persistence of consciousness among prominent EEG low-frequency oscillations (i.e., <4 Hz) has been observed in peculiar genetic conditions, i.e., Angelman syndrome [43], in liminal, transient, disconnected psychedelic states [154], and, occasionally, across the vegetative state, deep sedation, and NREM sleep [44,51,121,123–125].

Interestingly, studies during NREM sleep have recently identified two distinct types of slow waves, whose occurrence is anticorrelated, changes with sleep depth, and is differentially related to reports of conscious experience [50,155]. Type I slow waves are large, steeply rising, isolated events that occur more frequently in lighter NREM sleep; they exhibit a broad, synchronous cortical distribution and are followed by fast oscillations. Resembling the K-complex, these waves are thought to reflect diffuse subcortical arousing inputs from brainstem and thalamus and are associated with dream reports in within-state paradigms. Conversely, type II slow waves are small, shallow, frequent focal oscillations that become relatively more prevalent in deep NREM sleep; they exhibit variable origin and propagation and are not followed by fast oscillations. These waves are thought to be generated by local cortico-cortical interactions and, crucially, their presence predicts reports of no conscious experience. Although the electrophysiological features of post-hemispherotomy slow waves (Figs 2 and 4, and M and O in S1 Appendix)—alongside their inherent disconnection from subcortical influences—is consistent with type II slow waves, further confirmation with high-density EEG recordings may strengthen the inference about the potential absence of dream-like experience.

Further, the slowing observed at the scalp level should be characterized with intracranial recordings—in cases in which clinical outcomes require postoperative invasive monitoring. Such exploration would be very informative in this context because slow waves recorded at the scalp level can mask the presence of specific regional patterns that may be relevant for consciousness. For example, intracranial recordings have shown that local wakefulness-like intrusions can occur during NREM parasomnias, remaining largely undetected by scalp EEG [156,157]. In contrast, a recent case report involving the surgical isolation of a portion of the frontal lobe [158] revealed an EEG dominated by slow waves, along with a localized intracranial pattern of burst suppression—a hallmark of the deepest stages of anesthesia and profound coma [117,159–161]. Likewise, perturbational approaches that combine transcranial magnetic stimulation (TMS) and EEG [162,163], which are highly sensitive to detecting consciousness even in the presence of strong delta activity [164,165], may provide further insights. In cases where olfactory pathways to the isolated cortex remain intact, it will be also important to assess whether higher-level processes, such as the discrimination between familiar and unfamiliar odors, are preserved [12,166]. Finally, additional EEG-based markers of consciousness could be employed [63,167–169] and assessed against an appropriate normative reference, to further characterize the state of the isolated cortex.

## Conclusions

Our findings demonstrate that hemispherotomy induces a lasting reorganization of neurophysiological activity in the isolated cortex toward a state characterized by prominent slow waves and a marked broadband spectral redistribution toward lower frequencies. Characterizing this state by the Spectral Exponent, we detected values aligned to those observed during pediatric deep NREM sleep, indicative of a unihemispheric sleep-like state during wakefulness. The overall pattern is consistent with a state of absent or reduced consciousness. The persistence of slow waves, lasting months to years after surgery, raises the question of whether they play any functional role or merely reflect a regression to a default mode of cortical activity.

## Supporting information

**S1 Appendix. Table A.** Demographic information for each patient that underwent cortical disconnection of a single entire hemisphere (hemispherotomy), or of the parieto-occipito-temporal area (hereby noted as posterior), lateralized to one hemisphere. **SA**—Exclusion of EEG segments due to epileptic activity. **Fig A.** Excluded or included data segments due to epileptic activity. **SB**—The isolated cortex shows an increase in total power, in the context of a broad-band power redistribution. **Fig B.** Surgical disconnection increases overall total EEG power in the isolated hemisphere with respect to the contralateral hemisphere after surgery and to the state of the isolated cortex prior surgery. **SC**—Fitting of the Spectral Exponent and supplementary results. **SC.1**—Fitting of the Spectral Exponent. **Fig C.** Residuals/Periodic Component and Proportion of channels excluded from fit. **Fig D.** Group-level broad-band PSD and aperiodic trend in hemispherotomy, estimated over a broader 0.5–40 Hz range. **SC.2**—Results in the patient dataset of the Spectral Exponent, estimated between 1 and 20 Hz. **Fig E.** Replication of the findings of Fig 2 in the patient dataset on the Spectral Exponent, estimated over the 1–20 Hz range. **SD**—Mixed effects models. **Table B.** ANOVA of mixed effect models reveal a significant interaction between time and spatial side, indexing that the surgical disconnection differently affected each hemisphere. **Table C.** Pairwise Contrasts across spatial and temporal conditions on EEG slowing features. **SE**—Comparing narrow-band versus broad-band slowing in cortical isolation. **Fig F.** Bootstrap analysis revealed that the Spectral Exponent showed stronger statistical effects than Slow Delta Power in detecting the electrophysiological effect of cortical isolation on the disconnected hemisphere. **SF**—Controlling for Etiology and Type of Hemispherotomy. **SF.1**—Etiology. **SF.2**—Type of hemispherotomy. **Fig G.** Narrow- and broad-band spectral changes after surgery reveal a marked EEG slowing of the isolated cortex, robust across etiology and type of hemispherotomy. **SF.3**—Comparing EEG slowing after cortical disconnection between patients with whole Hemispherotomy and those with Temporo-Parieto-Occipital disconnection. **SF.4**—Mixed effect models including "type of hemispherotomy" as a covariate. **SG**—Comparison of Spectral Exponent with the reference sample distribution of wakefulness and NREM sleep. **Fig H.** Comparing the Spectral Exponent in hemispherotomy and in a reference pediatric sample across wakefulness and sleep revealed that the values of the isolated hemisphere are compatible with those of deep NREM sleep, between N2 and N3. **SH**—Correlation analysis between age and EEG features among hemispherotomy patients. **Fig I.** Correlation analysis among hemispherotomy patients revealed the absence of significant correlations between age and EEG features, either Slow Delta Power or Spectral Exponent. **SI**—Supporting age-matched comparison of Spectral Exponent between Patient and Reference datasets. **Fig J.** Matching subjects by age across the Patient and the Reference dataset. **Fig K.** Age-matched analysis comparing Spectral Exponent values in hemispherotomy to the pediatric reference dataset across wakefulness and NREM vigilance stages. **SJ**—Slow Delta Amplitude Envelope. **Fig L.** Estimation of the amplitude envelope in the Slow Delta frequency band (0.5–2 Hz). **Fig M.** Following surgery, the disconnected cortex displays a higher amplitude envelope of Slow Delta oscillations with respect to those observed in physiological wakefulness, yet smaller than that observed during physiological NREM sleep. **SJ.1**—Slow Delta PSD. **SK**—Slow Delta Period. **Fig N.** Following surgery, the disconnected cortex displays a longer period of Slow Delta oscillations with respect to those observed in both physiological wakefulness and NREM sleep. **SL**—Supporting

age-matched comparison of slow wave properties between Patient and Reference datasets. **Fig O.** Age-matched analysis of amplitude and period of Slow Delta oscillations in the disconnected cortex compared to the reference pediatric dataset during wakefulness and NREM sleep. **SM**—Regional aspects of EEG features among hemispherotomy patients. **Fig P.** Regional analysis of Slow Delta Power (0.5–2 Hz) following whole-hemisphere hemispherotomy revealed unihemispheric slowing after surgery, across anterior and posterior regions. **SM.1**—Postero-Anterior specificities of Slow Delta Amplitude. **Fig Q.** Regional age-matched analysis of amplitude and period of Slow Delta oscillations following whole-hemisphere hemispherotomy ($N = 7$) in relation to the pediatric reference dataset across wakefulness and NREM sleep. **SM.2**—Postero-Anterior specificities of Slow Delta Period. **SN**—Sigma oscillations and spindle activity. **SN.1**—Power of Sigma Periodic activity. **Fig R.** Analysis of total and periodic PSD in the sigma band reveal the absence of rhythmic spindle activity in the disconnected cortex. **SN.2**—Sigma burst detection. **Fig S.** Schematic depiction of the method employed for sigma bursts detection in a sleep recording, and observation of wakefulness-like sigma burst properties in the disconnected cortex. **SN.3**—Absence of spindles in the disconnected cortex.
(DOCX)

## Acknowledgments

Thanks to all the children and their parents who made this work possible. Thanks to the hospital technicians who performed the standard EEG recordings. Jacopo Favaro thanks the TMO group for inspiring discussions and the constant support. Thanks to Andrea Pigorini and Sasha D'Ambrosio for providing ideas for the cover image. Thanks to Gianluca Gaglioti for valuable insights and discussions.

## Author contributions

**Conceptualization:** Michele Angelo Colombo, Jacopo Favaro, Ezequiel Mikulan, Simone Sarasso, Tim Bayne, Anil K. Seth, Marcello Massimini.

**Data curation:** Jacopo Favaro, Andrea Pigorini, Flavia Maria Zauli, Ivana Sartori, Piergiorgio d'Orio.

**Formal analysis:** Michele Angelo Colombo, Ezequiel Mikulan.

**Funding acquisition:** Marcello Massimini.

**Investigation:** Flavia Maria Zauli, Ivana Sartori, Piergiorgio d'Orio, Laura Castana.

**Methodology:** Michele Angelo Colombo, Ezequiel Mikulan, Andrea Pigorini, Simone Sarasso.

**Project administration:** Marcello Massimini.

**Resources:** Andrea Pigorini, Flavia Maria Zauli, Ivana Sartori, Piergiorgio d'Orio, Laura Castana.

**Software:** Michele Angelo Colombo, Ezequiel Mikulan.

**Supervision:** Irene Toldo, Stefano Sartori, Simone Sarasso, Tim Bayne, Anil K. Seth, Marcello Massimini.

**Validation:** Michele Angelo Colombo, Jacopo Favaro.

**Visualization:** Michele Angelo Colombo, Jacopo Favaro, Ezequiel Mikulan, Andrea Pigorini.

**Writing – original draft:** Michele Angelo Colombo, Jacopo Favaro, Marcello Massimini.

**Writing – review & editing:** Michele Angelo Colombo, Jacopo Favaro, Ezequiel Mikulan, Andrea Pigorini, Irene Toldo, Stefano Sartori, Simone Sarasso, Tim Bayne, Anil K. Seth, Marcello Massimini.

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
