## [Editor Report · Decision Letter 0]

5 Feb 2025

Dear Dr Favaro, 

Thank you for submitting your manuscript entitled "Hemispherotomy: a cortical island of sleep-like activity in awake humans" for consideration as a Research Article by PLOS Biology.

Your manuscript has now been evaluated by the PLOS Biology editorial staff as well as by an academic editor with relevant expertise and I am writing to let you know that we would like to send your submission out for external peer review.

Once your full submission is complete, your paper will undergo a series of checks in preparation for peer review. After your manuscript has passed the checks it will be sent out for review. To provide the metadata for your submission, please Login to Editorial Manager (https://www.editorialmanager.com/pbiology) within two working days, i.e. by Feb 07 2025 11:59PM.

Kind regards,

Christian

Christian Schnell, PhD

Senior Editor

PLOS Biology

cschnell@plos.org

---

## [Decision Letter · Decision Letter 1]

4 Mar 2025

Dear Dr Massimini,

Thank you for your patience while your manuscript "Hemispherotomy: a cortical island of sleep-like activity in awake humans" was peer-reviewed at PLOS Biology. It has now been evaluated by the PLOS Biology editors, an Academic Editor with relevant expertise, and by several independent reviewers. 

In light of the reviews, which you will find at the end of this email, we would like to invite you to revise the work to thoroughly address the reviewers' reports.

As you will see below, the reviewers are overall supportive of publishing your study, but raise a few concerns that need to be addressed. In particular, they recommend (i) that the sections on sleep spindles are reworked or possibly removed completely, (ii) to expand on the significance of these findings within the broader context of consciousness science, and (iii) to provide detailed analyses and report on the dynamics of high frequencies.

Given the extent of revision needed, we cannot make a decision about publication until we have seen the revised manuscript and your response to the reviewers' comments. Your revised manuscript is likely to be sent for further evaluation by all or a subset of the reviewers.

**IMPORTANT - SUBMITTING YOUR REVISION**

*Re-submission Checklist*

*Published Peer Review*

*PLOS Data Policy*

*Blot and Gel Data Policy*

Sincerely,

Christian

Christian Schnell, PhD

Senior Editor

PLOS Biology

cschnell@plos.org

REVIEWS:

Reviewer #1 (Bjørn Erik Juel): Review of "Hemispherotomy: a cortical island of sleep-like activity in awake humans" by Colombo et al (2025)

General comment and recommendation

This work by Colombo, Favaro and colleagues presents an interesting investigation of a fascinating patient population, and asks a timely question: Do we have reason to believe a disconnected hemisphere may support consciousness despite being isolated from the world (including the rest of the body and brain)? The authors use methods and measures that are as well suited to answer the question as any we currently have available, and carry out their analyses meticulously. Their results are well presented and the article (largely) is written in an understandable and straightforward manner. That said, I find the conclusions drawn and the interpretations provided to be lacking. They seem to take their findings to be evidence against the presence of consciousness, while it seems a more prudent conclusion is that it does not provide (strong) evidence for the presence of (wakelike, rich, vivid, narrative etc) consciousness we are used to in normal waking life. To support their conclusion, they repeatedly claim that conditions such as sleep and anesthesia are unconscious, even though these are well known to be associated with dreams (albeit often vague or strange ones). Furthermore, the manuscript seems lacking in its presentation of the EEG literature on the relevant patient population and appears to present itself as the first of its kind. While it may very well be novel in the particular question it poses, it is by no means the first of its kind more broadly speaking wrt the methodology and patient population. In sum, I think the manuscript definitely deserves a chance of being published, but I strongly recommend rather extensive revision. 

In particular, 1) the introduction needs to more properly cover the literature on how EEG is affected by epilepsy and hemispherotomy (independent of whether the comparisons are done within subjects from pre to post surgery), 2) alternative measures of consciousness should be discussed and justifications for the choice made here should be improved, 3) the (pre-stated) hypotheses must be more concretely formalized to give a better understanding of which analyses were planned ahead of seeing the data, 4) a more proper discussion of the presence of conscious experiences in (so-called) "unconscious" states must be provided and used to contextualize what we can say about the presence of "islands of awareness" in this population (and a concrete definition of this phrase must be given—are dreams sufficient?), 5) a reevaluation of the need for the "spindle analysis" is required, as it currently takes up too much space for my liking, and 6) the phrasing of what this study actually gives evidence for (and not) must be improved (i.e. is this evidence AGAINST, or LACK of evidence FOR, islands of awareness (and why)?). 

Below are all the comments I made for myself while reading through the manuscript. While all of them do not need explicit answers, they are hopefully useful for guiding the authors in their revisions. Best of luck!

Title and highlights

The colon title makes it seem as though Hemispherotomy somehow is a cortical island. This is clearly not the intent. Perhaps adding a phrase like "evidence for" after the colon would help?

The emphasis on "sleep-like", which is then immediately undermined by the lack of spindles etc seems odd to put in the title. Maybe rather emphasize slow waves or the steeper spectral slope? 

I'd recommend reworking the title. 

Abstract

Authors state that they explore "for the first time the electrophysiological state of the isolated cortex before and after [hemispherotomy]". This does not seem to be the case. A quick google search yielded what looked like many studies employing similar strategies (although not to answer the same question). E.g The significance of bilateral EEG abnormalities before and after hemispherectomy in children with unilateral major hemisphere lesions - PubMed and EEG lateralization and seizure outcome following peri-insular hemispherotomy for pediatric hemispheric epilepsy | Child's Nervous System. These are just two examples, from a fast search. I urge the authors to properly review the literature on this. I may well be misunderstanding, and be mistaken in that the author's claim is false, but it seem very surprising to me that no studies comparing EEG before and after hemispherotomy have ever been published. 

This statement seems imprecise: " a broad-band shift in power spectral density from high to low frequencies." I assume the intent is to show the change in slope. But looking at the PSDs, the claim does not seem supported. There is an increase in low delta, and that's it. Consider rephrasing to explicitly mentioning the spectral slope. 

This claim is too strong: "challenging the possibility that hemispherotomy might lead to inaccessible "islands of awareness." Given that conditions with comparable slopes and EEG characteristics are commonly associated with dreams, there may very well still be "islands of awareness" (insofar as "awareness" is not taken to strictly be referring to conscious experiences of external stimuli or "process" data coming in from outside). 

Introduction 

You say, " More broadly, the activity patterns that extensive regions of the human brain can sustain following massive deafferentation remain unknown", but I feel some things Can be assumed given work on isolated cortical slices (e.g timofeev et al 2000 and Sanchez-Vives and McCormick 2000). I feel the assumption that slowing might be expected (e.g. given the senior author's recent perspective "Shaping the Default Activity Pattern of the Cortical Network") should be briefly discussed or introduced here to help the reader understand the context, and choices of measures applied. 

The introduction goes on to claim that there is little to no relevant studies using EEG. I find this to be strange, as there is a long tradition of using EEG to monitor epilepsy patients, both for diagnosis, localization of seizure foci, prognosis for surgery, and evaluation of effectiveness of surgery. This literature needs to be covered in the introduction, or a strong argument for why it can be bypassed needs to be presented. 

The paragraph starting with "After hemispherotomy we observed…" seems to be a mix of results and discussion, before going back to justifying methodological choices without first clearly stating the aims and hypotheses of the study. While I understand what the authors are doing in this study, I think the introduction could be greatly improved. In addition to the missing discussion of EEG-studies in relevant patient populations, I think there should be a better introduction of available methods for assessing consciousness objectively using EEG, providing strong justification for why the two main measures were chosen. 

In particular, the introduction suffers from a lack of clear statements of aim/hypotheses. Given this, it is unclear what were the pre-planned analyses, and which were descriptive/post-hoc. For example, the claim "We accordingly quantified slow wave activity by narrow band spectral analysis within the SlowDelta band (0.5-2 Hz)," indicates that it was preplanned, but i get the feeling (from looking at the figures) that this range is rather post-selected since it is the only region with significant differences. Please specify and make explicit what the concrete hypotheses were (prior to looking at the data). 

Methods 

You say "We retrospectively selected our participants". This, along with the inclusion criteria, indicates that all recordings were already done, but it is not obvious (as you also say "Ten participants were recruited"). Please be a bit more specific about this. I need to be spoon fed :)

You say "which required relatively high spatial specificity". What do you mean by this? Please clarify. 

There is a notable change in the narrative structure of the methods in the "Sigma periodic activity" section. That is, there is much more background and justification for the analysis than in the rest of the methods. In particular, given that there is little reason to believe there will be spindle activity in the disconnected cortex (arent spindles largely driven by thalamic activity?), it seems rather lopsided. Please reconsider whether this analysis deserves so much space. You also go on to repeat much of the justification and explanation in the results section. 

Results

The large range of ages in the sample is a bit worrisome, given that there are large changes in EEG patterns that happen in this age range (see for example the review by Gorgoni et al (2020) in Sleep Medicine). Consider providing some supplementary figures showing how the key markers used here (and differences in them) are influenced by age. 

Is the 11 year old girl really "A representative example?" Especially given that it is one of the older patients?

I am a bit confused as to why fig 2 is not given more space in the results section, as it seems to be the main figure/result pertinent to the central thesis of the paper. 

Figures

Fig 1 and 2. The slopes may be significantly different, but the differences in the PSD are vastly smaller than those seen the original paper by Colombo et al. proposing the spectral slope as a measure of consciousness. E.g. the only significant difference in absolute powers is in the low delta range. This could be explained by the difference in age/recording equipment etc, but to me the rather minor changes in the actual PSD make me distrust strong claims about what we can say about the awareness of the underlying hemisphere. Just eye-balling it, the changes in the 1-20 hz seen in this study resemble more the changes previously reported to the ketamine state (see this from colombo et al. although, of course, I understand that "eye-balling" is insufficient :)). 

Figure2: Why is the statistical comparison swapped here, relative to fig 1? Why is not the pre vs post different directly shown in the figure? I recommend consistency 

Fig 5: I recommend this be "demoted" to supplements, preferably together with the majority of the spindle discussion (as it seems more confirmatory of surgery success than anything else—im happy to be corrected on this point!)

Discussion

The first sentence in the discussion is not correct. There are many studies (the earliest I found from a quick google search was from 1950) investigating how EEG changes following hemispherectomy in humans. Please edit to make the novel aspects of this study correctly reflected. The easiest would (probably) be to emphasize the context of consciousness studies. Or just remove it—no need for salesmanship. 

Also in the first paragraph of the discussion, i think the following statement needs revision: "Such broad-band neurophysiological slowing, indexed by the spectral exponent, was similar to that previously found in the physiological [...], pharmacological [...] and pathological unconscious states." In particular, I think it is questionable to what extent changes seen here were "similar" (although the direction of change was the same) and i would be hesitant to calling them "unconscious states", as these states are all riddled with (albeit arguably "reduced" and disconnected) conscious experiences. 

Without also noting that the "pre surgery state" in the disconnected hemisphere resembled N1 sleep, I feel the following is over-selling: "Upon direct comparison with a reference pediatric sample, the disconnected cortex showed a broad-band spectral exponent resembling deep NREM sleep." 

I also, again, fail to see the relevance (or need for emphasis) of the following sentence, given that the hemisphere is disconnected from subcortical structures (or am I misunderstanding?). If anything, this seems to me like a basic confirmation that the surgery was successful, but I fail to see the reason for why it requires so much space throughout. 

You could add the Wada test to this: "A vast literature on acquired brain-injury [...], hemispherectomy [...] and split-brain patients [...] suggests that one hemisphere is sufficient to support consciousness." Perhaps also the fact that several animals (e.g. migratory birds and dolphins) can have half their brain asleep while remaining vigilant and behaviorally awake. 

I urge you to avoid this formulation: "unconscious states such as sleep and anesthesia". Both of these states are well known to have a lot of dreams, despite it being difficult to obtain reports of rich reports (for different reasons). Even in deep sleep, and surgical anesthesia, there are reports of dream reports (albeit, arguably, of less rich experiences than normal wakefulness) in a large percentage of awakenings. 

Unfortunately, I have to disagree with the following statement: "the disconnected cortex rests in an electrophysiological state that is not compatible with the presence of an island of awareness". There are many situations where the cortex is in a state of slow waves where there is provably (to the extent that it is possible, relying on immediate reports following abrupt awakening). This must be weakened, or rephrased. As a minimal change (which changes the emphasis of what we can get evidence for from observational studies such as this one) you might say something like "the disconnected cortex rests in an electrophysiological state that does not provide strong evidence for the presence of an island of awareness". Note the switch from evidence against consciousness to lack of evidence for consciousness.

Unsurprisingly, I think the following is much to strongly phrased: "First, EEG slow waves are canonically associated with unconscious conditions encompassing NREM sleep[...], general anesthesia[...] and the vegetative state [...]." Again, all of these states are well known to be associated with conscious experiences.

The following interpretation seems more valid, but is in contrast with the stronger claims brought elsewhere: "indication of the reduced likelihood of dream-like experiences in the isolated cortex." 

Again, I think the following has the wrong emphasis: "provide evidence against the possibility that hemispherotomy may result in cortical islands of awareness." It is not that there is evidence against, but (at best) no evidence for, vivid/rich/wakelike/narrative experiences. Better would be something like saying the likelihood of receiving reports of experience (if it could give such reports) would be reduced. Meaning that there is still a (high!) probability that there are experiences, but that the kind and frequency would be more like those from deep sleep, anesthesia etc. 

The paragraph starting with "Crucially, the disconnected cortex" nicely sums up my issue with that analysis. First, the "crucially" indicates that this is an important finding to understand/interpret the rest of the findings, even though no justification for this is given. Then it ends with the sentence "This result is unsurprising, as spindles are generated in the thalamus[...]" which undermines the need for this analysis. Again, I recommend reassessing the need for this analysis, beyond just confirming the success of the surgery. To me, it just muddles the paper. 

I think this is too strong: "aligns with patterns observed in unconscious reference conditions, alleviating concerns about the presence of an "island of awareness" within the disconnected hemisphere." First, there is no direct evidence that the reference conditions were unconscious (a large proportion of dreams even in N3). Second, the extent to which it "aligns" is debatable, in part also, because of what you discuss earlier: pre surgery the disconnected hemisphere was like N1, without that leading us to believe there were N1-like experiences at that point. Third, at least to me, it doesnt "alleviate" much if anything wrt the possibility of islands of awareness. It does, perhaps, indicate that the experiences it might be having are more like those we commonly see in N2 sleep, deep propofol sedation, and the like (that is, dreams in upwards of 80% of awakenings, depending on methods for obtaining reports), but is that really evidence for unconsciousness and no islands of awareness? I dont think so. 

The discussion immediately following this is better, but does not touch on the fact that most conditions classically (and incorrectly) assumed to be "unconscious" states, are in fact (as far as we can tell with gold standard approaches based on immediate reports) riddled with conscious experience. This should be made very explicit in this discussion, rather than just pointing to some rare genetic conditions. This is because it provides important context wrt drawing conclusions about whether there is good reason to believe there can be any awareness (or, rather, experiences) in the disconnected cortical islands

I am missing a conclusion of the overall, main findings of the study to wrap up the discussion section. 

Reviewer #2: The study "Hemispherotomy: a cortical island of sleep-like activity in aware humans" by Colombo and colleagues subjects ten pediatric hemispherotomy patients to pre- and postoperative EEG-analysis. While a myriad of research studies have analyzed preoperative clinical, psychological, electrophysiological and imaging predictors in hemispherotomy candidates (Staudt et al., 2002; Gaubatz et al. 2020; Ramantani et al., 2024 and others), the fate of the isolated hemisphere has not received scientific attention. This is interesting as subjects who underwent the (less invasive) callosotomy have been in the center of decade-long neuroscientific and neurophilosophical research endeavours (Nagel, 1974; Wolman, 2012 and others). Only recently, the isolated hemisphere of individuals after hemispherotomy has been re-introduced in the debate by one of the study's coauthors (Bayne, 2020), when their capacity to be "islands of awareness" has been suggested. This most interesting claim, however, could not be substantiated due to a lack of empirical data. These data are now provided in this most spectacular study. This is a concisely written manuscript involving a very rare study collective, longitudinal data and most elaborate EEG-analysis. It finds large-amplitude slow wave activity and an overall redistribution of power spectral density from high to low frequencies in the isolated hemisphere. Interestingly, a lack of spindle and slow-delta activity with smaller amplitude is found, indicative of disconnected thalamo-cortical connections. The EEG-findings resemble sleep and, thus, question the claim that the isolated hemisphere after hemispherotomy holds a potential to be an island of awareness. The emerging scientific debate is being steered in a new direction by these results, for which the authors deserve great credit. I had several minor revisions after reading the study, only to find that they had almost all been answered by the authors in the most extensive supplementary material. The study's limitations are almost completely addressed in the limitations-and-future-directions-paragraph of the manuscript (such as the role of the pathological hemisphere in the constitution of consciousness before surgery). An additional limitation in all hemispherotomy studies is the clinical heterogeneity of the individuals regarding etiology, age at surgery and type of surgery. While authors provide subgroup analysis and a mixed effects model in the supplement, they have not used the lesion volume and location as a covariate in their analysis which would have been of interest. 

Reviewer #3 (Igor Timofeev): In this study Colombo et al investigated EEG activities in young human epileptic patients before and after complete isolation of cortex or large part of cortex in one hemisphere that was responsible for the seizure onsets. The authors report that isolated large parts of the cortex show continuous slow wave activities that are reminiscent, but not identical to slow wave activities recorded during slow wave sleep. Based on previous animal experiments, such a finding could be expected but it was never demonstrated in human. Therefore, this study provides fundamental new information on electrographic behavior of isolated human cortex. Bravo. 

Despite my great enthusiasm about this study, I have several general comments which, if addressed, would likely improve the presentation of the results, and I have also some specific/minor comments which point to some small errors or unclearly described aspects of the manuscript.

General comments. 

1. Essentially you compared EEG in a 'pathological' hemisphere before and after disconnection with contralateral hemisphere without specifying in each case the state of vigilance with control population in which you specify W, N1, N2 and SWS. I was continuously looking for a vigilance state in your patient population. So, ideally, you should present data in your patient group with specified W, N1, N2 and SWS. If you do not have such data because such recordings were clinically not justified, I would still recommend mentioning in each figure legend that your patient recordings were done during wake. More, if all your recordings in patients were done in wake, than a part of your study related spindle analysis has no meaning because spindles during wake are not generated. Then, I would strongly recommend removing the part of your manuscript related to spindles. With or without isolation, spindles would not be generated.

2. All your data presented as average across all EEG electrodes and patients. This give you higher statistical power, but there are some drawbacks too. (a) It is known that in 2-year-old children that slow wave activity is highest in the occipital cortex and in 10-year-old children the highest power is in central and frontal regions (Kurth S, Ringli M, Geiger A, LeBourgeois M, Jenni OG, Huber R. Mapping of cortical activity in the first two decades of life: a high-density sleep electroencephalogram study. J Neurosci. 2010; 30 (40): 13211-13219.). (b) As you previously demonstrated the sleep slow waves propagate. There are also animal data that cortical slow wave synchrony after isolation or thalamic inactivation is reduced thus propagation velocity is reduced (you cited Lemieux et al., 2014, but they also have follow up study on that topic (Lemieux M, Chauvette S, Timofeev I. Neocortical inhibitory activities and long-range afferents contribute to the synchronous onset of silent states of the neocortical slow oscillation. J Neurophysiol. 2015; 113 (3): 768-779). Based on these two facts, your conclusions on properties of slow waves in isolated cortex could be erroneous. If you average slow waves that propagate slower in one case as compared to control case, or propagate from posterior to anterior or central to peripheral regions without known velocity, you can obtain averaged data showing smaller slow wave amplitudes, different periodicity, etc. Since you have only 10 patients from 2.5 to almost 12 years of age, it would be difficult to make groups, but you can do plots showing measured parameters of slow waves of individual patients on one axes and age on a second axes. To address the issue of regional specificity, you can average data obtained over a few electrodes located over frontal lobe vs. occipital lobe etc. If you would do such or similar analysis and obtain information that is congruent with your current findings, it will reinforce your point regarding slow wave properties. If by whatever reason, you cannot do such analysis, you can add a brief paragraph in you limitation section describing this as a limitation.

Specific/minor comments

It would be important to describe briefly paroxysmal activities in the isolated cortex, in particular, because such activities were previously described in human isolated cortex (Echlin FA, Arnet V, Zoll J. Paroxysmal high voltage discharges from isolated and partially isolated human and animal cerebral cortex. Electroencephalography and Clinical Neurophsysiology. 1952; 4: 147-164.). 

In multiple figures it is unclear which state is reported. Some time it is very confusing. iIn Fig. 3 legend you state 'The contralateral cortex, both before and after surgery, showed values compatible with wakefulness'. Were patients awake or they were asleep but values of spectral exponent were compatible with wakefulness.

p. 17 of pdf …reached all NREM sleep stages (up to N3),… What classification of states was used? In (Iber C, Ancoli-Israel S, Chesson A, Quan S. Manual for the Scoring of Sleep and Associated Events: Rules, Terminology and Technical Specifications. Westchester: American Academy of Sleep Medicine; 2007) there is stage 1, stage 2 and SWS. Stage 3 was used in earlier classifications.

p. 17 of pdf. …virtual EEG reference… unclear to me what is "virtual EEG reference"? Please explain.

The calculation of Amplitude of Slow-Delta activity is unclear to me. Could you please make a panel in Suppl. Fig. 6 to explain this graphically?

In the section entitled 'Period of slow delta oscillations and rate of large slow waves' I did not find how you defined and detected 'large slow waves'?

Please be consistent with terminology/wording. In the section 'Sigma oscillations and spindle activity' you talk about low amplitude alpha activity, but no word about sigma! In the main text, regarding spindle detection you write you relayed on a trained neurologist, which is not reliable to me for such a fine study, but in the supplemental materials, you describe a reliable automatic method of spindle detection. 

The first paragraph of the method section and the first paragraph of the result section largely repetitive.

Fig. 1. It is my understanding that all the data in this fig are coming from a single subject. It would be nice to indicate this in the title of the fig. 

Fig. 1B. I presume that recordings in the right and left hemispheres (pre and post) are done simultaneously. It is not explicitly stated.

Fig. 1 C and D. Several things are unclear: (1) What was the state NREM sleep (stage?), REM sleep or wake?, (2) What was the duration of the analyzed and shown recording?

Fig. 2 B-E Square symbols and dots are too small to be visible.

Fig. 3 and supplementary fig. 5 upper part looks to be identical. Such a repetition is not needed. However, it would be important to see in patient data set dynamics of slow/delta power and or spectral exponent throughout states: W, N1, N2 and SWS before and after disconnection, something like what you have shown for the reference dataset. The bootstrapped differences analysis is going to this point, but it looks a bit confusing to me. Either it shows something that I do not understand, or the description is not adequate. Ideally, you should present real measurements.

Suppl Figs 6 and 7 please increase font size. It is almost impossible to read with the current font size.

Suppl Figs 6 and 7 has panels identical to the panels shown in the fig 4. Fig. 4 is not mentioned in the result section.

Fig. 4, Suppl Fig 7 and related text. You mention here slow/delta waves during wake in contralateral cortex. The presence of slow waves during wake (unless patients were sleep deprived) is a big surprise to me. Please illustrate with original recordings the slow/delta waves presence during wake.

Fig. 5. Please increase font size. I cannot read any numbers in the insets even after magnification.

Fig. 5 and related text. I have again a problem with your data presentation. For your reference group all is clear. During sleep there is a maximum around 13 Hz, which likely represent spindles. In your patient group, the state of vigilance is not indicated and in contralateral as well as in pathological hemisphere, there is no sign of a peak around 13 Hz. The peak is around 9 Hz, likely alpha. This indicates to me that the recordings were collected during wake. So how did you expect to record spindles during wake? Why do you make a section of your paper about spindles in a condition in which spindles cannot be observed? Please make sure that you show data separately for NREM sleep and wake.

p. 29 of pdf. '… longer periods compared to that seen during physiological sleep (Figure 5).' You likely mean Fig. 4. Fig. 5 is about spindles.

p. 30 of pdf '…consistent with the characteristics of physiological wakefulness (Figure 4,…' Do you mean Fig. 5 or some supplemental figs?

p. 30 of pdf '…interesting questions about the role of slow waves after disconnection.' It seems that there is no role of slow waves in isolated cortex. As you mentioned above, slow wave activity in isolated cortex is default state.

---

## [Decision Letter · Decision Letter 2]

8 Jul 2025

Dear Dr Massimini,

Thank you for your patience while we considered your revised manuscript "Hemispherotomy: a cortical island of sleep-like slow waves in awake humans" for consideration as a Research Article at PLOS Biology. Your revised study has now been evaluated by the PLOS Biology editors, the Academic Editor and the original reviewers. 

In light of the reviews, which you will find at the end of this email, we are pleased to offer you the opportunity to address the remaining points from Reviewer 1 and Reviewer 3 in a revision that we anticipate should not take you very long. We will then assess your revised manuscript and your response to the reviewers' comments with our Academic Editor aiming to avoid further rounds of peer-review, although we might need to consult with the reviewers, depending on the nature of the revisions.

**IMPORTANT - SUBMITTING YOUR REVISION**

*Resubmission Checklist*

*Published Peer Review*

*PLOS Data Policy*

*Blot and Gel Data Policy*

Sincerely,

Christian

Christian Schnell, PhD

Senior Editor

PLOS Biology

cschnell@plos.org

Reviewer #1 (Bjørn Erik Juel signed his report): First of all, I would like to commend the authors on a tremendous effort in the revision process. Despite the largely positive reviewer feedback, the authors went above and beyond what I would typically expect in replying to my previous comments. In other words, I am satisfied with their effort, and accept almost all edits proposed in the resubmission. 

There is, however, one set of responses from the authors (with associated changes in the resubmission) that I cannot leave unanswered. It is most prominent, I think, in the response to point 5 (to reviewer 1). They propose the following sentence: "The findings of prominent slow oscillations and broad-band slowing support inferences of absent or reduced awareness in the isolated cortex." However it also shines through in other responses and edits, like "In sum, our findings of a unihemispheric electrophysiological spectral profile resembling deep-sleep are consistent with absent, or reduced conscious experiences, rather than constituting definitive evidence against any form of conscious experience" in combination with "It remains possible that some form of consciousness could persist even in the presence of highly prominent slow waves in the spontaneous scalp EEG, as revealed by peculiar genetic conditions, i.e. Angelman syndrome (Frohlich et al., 2021), and by some transient, disconnected psychedelic states (Blackburne et al., 2024). More broadly, growing evidence suggests that some form of disconnected, residual consciousness may persist in states traditionally labeled as "unconscious"—such as deep sedation, NREM sleep, and the vegetative state."

While I am well aware of the long standing tradition of labeling states such as nrem sleep and deep sedation as states of unconsciousness or absent of experience (or the weaker, but similar, "reduced consciousness"), it is just not the case that such states are typically devoid of experience, or even necessarily associated with something that is reasonably labeled with the diminutive "residual consciousness". To the contrary, reports of rich experiences are not that uncommon (from both NREM and propofol sedation (RASS -4/MOAAS 1), and combined with the well known amnesic effects of anesthetics, and the confused and tired state of individuals trying to report their experiences upon awakening from deep nrem sleep or sedation, it seems more reasonable to err on the side of caution in these situations. Of course, I understand that it is sometimes necessary to provide simple stories and narratives (e.g. in clinical situations where next-of-kin might need comfort), but an academic article is not the place for it. 

To deal with this, I recommend something like the following: 

- In place of phrases like "absent or reduced awareness" substitute "conscious experiences similar to those typical of NREM sleep and deep sedation". This maintains the possibility that the disconnected hemisphere has no, reduced, residual, non-rich experiences (depending on what is actually the case in NREM etc), without drawing assumptions that have a high likelihood of being too simplistic, given the current state of the evidence. 

- Alternatively, make it even more clear that the current evidence is, contrary to the traditional view, that NREM sleep, deep sedation, and other states with pronounced slow waves are generally not devoid of conscious experience (despite the problems associated with reporting such experiences), but that the authors nevertheless choose to assume that they are in fact best characterized as states with "absent or reduced awareness".

Finally, I would recommend removing speculation/inferences about conscious state/contents of the disconnected hemisphere from highlights and abstract, and leave it purely to the discussion as interpretation of the results (with a clear indication that inferences about absent/reduced/un consciousness are based on common but empirically disputed assumptions regarding the ground truth of consciousness in nrem/deep sedation/UWS/etc). Something similar, I think, should be the case for the end of the introduction: avoid making too strong assumptions about what slow-waves allow you to infer about (un)consciousness. Despite the authors claiming (with many references!) that the measures used is "validated as a reliable EEG marker of the level of consciousness across a range of physiological [...], pharmacological [...], and pathological states", it is, unfortunately, the case that the ground truth about absence of (or reduced) consciousness was rarely adequately controlled for in the tested populations/cited studies, and thus the success of the marker (and most others with it) relies on the traditional (but too simplistic) preconceived assumptions about presence/level of consciousness in those populations. (Im not saying there are other markers that fare any better, just that most studies to date of testing/validating markers of consciousness have taken for granted the traditional view of consciousness in nrem/sedation/uws/etc)

Anyways, I recognize that my (strong) opinions are countercultural and not typical of the traditional view of our field, and therefore I leave it up to the authors how/if they wish to take this points into consideration, and to the editor to make the final decision. In other words, I think the paper can easily be published as is, but I'll just reiterate that I think the paper would be more stringent and less speculative if empirically disputed, but traditionally common, assumptions about absent/reduced consciousness in slow-wave states were edited to be more agnostic about the ground truth.

Congratulations, and best of luck! 

Reviewer #2: The authors have answered all my questions satisfactorily, and I congratulate them on their work.

Reviewer #3 (Igor Timofeev signed his report): Overall, the authors provided satisfactory corrections to my critique. Therefore, I have only some minor suggestions.

Please check color code in Supplementary Fig. 3. I have feeling that pre and post colors, particularly for disconnect hemisphere are reversed.

Some supplementary figures are either not referenced in the text or they are not referenced in the order from Supp. Fig. 1 through Supp. Fig. 19. Please check and update.

p. 128 of PDF … with respect to physiological NREM sleep and wakefulness… Here, and in some other places below. It is clear that a major part of the study was done during waking state. Because still some recordings (reference dataset) were done during NREM, and in order to avoid any confusion, please double check that in each part of the manuscript and figure you clearly outline the exact state of the patient.

Supplementary Fig. 14. In the legend, you reference to panels A, B and C. In the figure itself the panels are not labelled as A, B, C. 

Some references in the text are not found in the list of references. I did not check all, but at least Lemieux et al., 2015 is cited in the text, but could not be found in the list of references.

---

## [Editor Report · Decision Letter 3]

30 Jul 2025

Dear Dr Massimini,

Thank you for your patience while we considered your revised manuscript "Hemispherotomy: a cortical island of sleep-like slow waves in awake humans" for publication as a Research Article at PLOS Biology. This revised version of your manuscript has been evaluated by the PLOS Biology editors and the Academic Editor.

Based on our Academic Editor's assessment of your revision, we are likely to accept this manuscript for publication, provided you satisfactorily address the following data and other policy-related requests:

* We would like to suggest a different title to improve its accessibility for our broad audience: 

Sleep-like neuronal oscillations persist in the isolated cortex of awake humans years after surgical separation of the brain hemispheres

* Please add the links to the funding agencies in the Financial Disclosure statement in the manuscript details.

* Please include the approval/license number of the ethical approval for the experiments.

DATA POLICY:

Regardless of the method selected, please ensure that you provide the individual numerical values that underlie the summary data displayed in the following figure panels as they are essential for readers to assess your analysis and to reproduce it: 1D, 2BCDE, 3, 4, S2, S5, S6, S7ABCD, S8, S11ABCDEFGH, S13 (right panel), S14 (right panel), S15ABCD, S16AB, S17ABCDEFGH, S18 (right panel) and 19BC.

* CODE POLICY

We expect to receive your revised manuscript within two weeks. 

*Published Peer Review History*

*Press*

Sincerely,

Christian

Christian Schnell, PhD

Senior Editor

cschnell@plos.org

PLOS Biology

---

## [Editor Report · Decision Letter 4]

29 Aug 2025

Dear Dr Massimini,

Thank you for the submission of your revised Research Article "Hemispherotomy leads to persistent sleep-like slow waves in the isolated cortex of awake humans" for publication in PLOS Biology. On behalf of my colleagues and the Academic Editor, Simon Hanslmayr, I am pleased to say that we can in principle accept your manuscript for publication, provided you address any remaining formatting and reporting issues. These will be detailed in an email you should receive within 2-3 business days from our colleagues in the journal operations team; no action is required from you until then. Please note that we will not be able to formally accept your manuscript and schedule it for publication until you have completed any requested changes.

PRESS

Sincerely, 

Christian

Christian Schnell, PhD

Senior Editor

PLOS Biology

cschnell@plos.org